# BREAKING NEURAL NETWORK SCALING LAWS WITH MODULARITY

## ABSTRACT

Modular neural networks outperform non-modular neural networks on tasks ranging from visual question answering to robotics. These performance improvements are thought to be due to modular networks' superior ability to model the compositional and combinatorial structure of real-world problems. However, a theoretical explanation of how modularity improves generalizability, and how to leverage task modularity while training networks remains elusive. Using recent theoretical progress in explaining neural network generalization, we investigate how the amount of training data required to generalize on a task varies with the intrinsic dimensionality of a task's input. We show theoretically that when applied to modularly-structured tasks, while non-modular networks require an exponential number of samples with task dimensionality, modular networks' sample complexity is independent of task dimensionality: modular networks can generalize in high dimensions. We then develop a novel learning rule for modular networks to exploit this advantage and empirically show the rule's improved generalization, both in and out of distribution, on high-dimensional, modular tasks.

## 1 INTRODUCTION

Modular neural network (NN) architectures have achieved impressive results in a variety of domains ranging from visual question answering (VQA) (Andreas et al., 2016a;b; Hu et al., 2017; Johnson et al., 2017; Yi et al., 2018; Kim et al., 2019), reinforcement learning (Goyal et al., 2021; Madan et al., 2021), robotics (Alet et al., 2018b; Pathak et al., 2019; Yang et al., 2020) and natural language processing for which modular architectures based on attention (Bahdanau et al., 2015) are standard. It is thought that modular NNs can generalize better by facilitating *combinatorial generalization*, a phenomenon where a learning system recombines previously learned components in novel ways to generalize to unseen task inputs (Alet et al., 2018b; D'Amario et al., 2021; Mittal et al., 2022; Jarvis et al., 2023). Yet, a fundamental understanding of why modularity benefits generalization is lacking.

In parallel, the generalization properties of *monolithic* (non-modular) NNs have been increasingly well understood both theoretically and empirically. In particular, current theory can explain the double descent phenomenon where NN generalization error decreases with increasingly large capacity (Belkin et al., 2019; Rocks & Mehta, 2022). NN learning in a certain regime can also be understood as kernel regression (Jacot et al., 2018). Extensive empirical studies have also measured scaling laws of NN generalization error (Kaplan et al., 2020), and moreover, these scaling laws can be explained theoretically (Bahri et al., 2021; Hutter, 2021). However, these laws indicate that the sample complexity required to generalize on a task scales *exponentially* with the intrinsic dimensionality of the task's input (McRae et al., 2020; Sharma & Kaplan, 2022). This raises the question: how can we hope to generalize on high-dimensional problems with limited training data?

In this work, we investigate how modular NNs can circumvent this exponential number of samples. We first synthesize existing generalization results in a unified theoretical model of NN generalization error and empirically validate it on tasks with varying intrinsic dimensionality. We then use our model to show theoretically that appropriately structured modular NNs avoid using an exponential number of samples on modular tasks. However, recent work shows that architectural modularity is in practice not sufficient on its own to solve modular tasks efficiently (Csordás et al., 2021; Mittal et al., 2022); a solution to align NN modules to a task's modularity is lacking. We propose a novel

learning rule that aligns NN modules to approach the underlying modules of a task and empirically demonstrate its improved generalization.

We summarize our contributions as follows:

- We propose a theoretical model of NN learning that synthesizes existing generalization results. Our model predicts the generalization error of NNs under variations of the number of model parameters, number of training samples, and dimensions of variation of a task. We empirically validate our theoretical model on a novel parametrically controllable sine wave regression task and show that sample complexity varies exponentially with task dimension.
- We apply the theoretical model to compute explicit, non-asymptotic expressions for generalization error in modular architectures; *to our knowledge, we are the first to do so.* Our result demonstrates that sample complexity is *independent* of task dimension for modular NNs applied to modular tasks of a specific form.
- Based on our theory, we develop a learning rule to align NN modules to the modules underlying high-dimensional modular tasks with the goal of promoting generalization on these tasks.
- We empirically validate the improved generalizability (both in and out-of-distribution) of our modular learning approach on parametrically controllable, high-dimensional tasks: sine wave regression and Compositional CIFAR-10.

## 2 RELATED WORK

### 2.1 MODULAR NEURAL NETWORKS

Recent efforts to model cognitive processes show that functional modules and compositional representations emerge after training on a task (Yang et al., 2019; Yamashita & Tani, 2008; Iyer et al., 2022). Partly inspired by this, recent works in AI propose using modular networks: networks composed of sparsely connected, reusable *modules* (Alet et al., 2018b;a; Chang et al., 2019; Chaudhry et al., 2020; Shazeer et al., 2017; Ashok et al., 2022; Yang et al., 2022; Sax et al., 2020; Pfeiffer et al., 2023). Empirically, modularity enhances out-of-distribution generalization (Bengio et al., 2020; Madan et al., 2021; Mittal et al., 2020; Jarvis et al., 2023), modular generative models are effective unsupervised learners (Parascandolo et al., 2018; Locatello et al., 2019) and modular architectures can be more interpretable (Agarwal et al., 2021). In addition, meta-learning algorithms can discover and learn the modules without prespecifying them (Chen et al., 2020; Sikka et al., 2020; Chitnis et al., 2019).

Recent empirical studies have investigated how modularity influences network performance and generalization. The degree of modularity increases systematic generalization performance in VQA tasks (D'Amario et al., 2021) and sequence-based tasks (Mittal et al., 2020). Rosenbaum et al. (2019) and Cui & Jaech (2020) study routing networks (Rosenbaum et al., 2018), a type of modular architecture, and identified several difficulties with training these architectures including training instability and module collapse. Csordás et al. (2021) and Mittal et al. (2022) extend this type of analysis to more general networks to show that NN modules often may not be optimally used to promote task performance despite having the potential to do so. These analyses are primarily empirical; by contrast, in our work, we aim to provide a theoretical basis for how modularity may improve generalization. Moreover, given that architectural modularity may not be sufficient to ensure generalization, we propose a learning rule designed to align NN modules to the modularity of the task.

### 2.2 NEURAL NETWORK SCALING LAWS

Many works present frameworks to quantify scaling laws that map a NN's parameter count or training dataset size to an estimated testing loss. Empirically and theoretically, these works find that testing loss scales as a power-law with respect to the dataset size and parameter count on well-trained NNs (Bahri et al., 2021; Rosenfeld et al., 2020), including transformer-based language models (Sharma & Kaplan, 2022; Clark et al., 2022; Tay et al., 2022).

Many prior works also conclude that generalizations of the power-law or non-power-law-based distributions can also well model neural scaling laws, in many cases better than vanilla power-law

frameworks (Mahmood et al., 2022; Alabdulmohsin et al., 2022). For instance, Hutter (2021) shows that countably-infinite parameter models closely follow non-power-law-based distributions under unbounded data complexity regimes. In another case, Sorscher et al. (2022) show that exponential scaling works better than power-law scaling if the testing loss is associated with a pruned dataset size, given a pruning metric that discards easy or hard examples under abundant or scarce data guarantees respectively.

Some works approach this problem by modeling NN learning as manifold or kernel regression. For example, McRae et al. (2020) considers regression on manifolds and concludes that sample complexity scales based on the intrinsic manifold dimension of the data. In another case, Canatar et al. (2021) draws correlations between the study of kernel regression to how infinite-width deep networks can generalize based on the size of the training dataset and the suitability of a particular kernel for a task.

Among other observations, this body of work shows that in the absence of strong inductive biases, high-dimensional tasks have sample complexity growing roughly exponentially with the intrinsic dimensionality of the data manifold. In this work, we investigate if learning the modular structure of modularly structured tasks will help in reducing sample complexity while training.

## 3 MODELING NEURAL NETWORK GENERALIZATION

In this section, we present a toy model of NN learning that treats NNs as linear functions of their parameters; this is along with the lines of prior work such as (Bahri et al., 2021; Canatar et al., 2021). Although this common theoretical assumption does not directly apply to practical, non-linear architectures, the assumption provides analytical tractability and, moreover, can be shown to predict generalization *even in nonlinear networks* (e.g. in the Neural Tangent Kernel literature (Jacot et al., 2018)). Under this setting, we find exact closed-form expressions for expected training and test loss even without taking asymptotic limits in the number of samples or model parameters. We find that our toy model captures key features of NN generalization applied to a sine wave regression task. We summarize our notation in Tab 1.

Table 1: Table of symbols.

| Symbol | Meaning |
|--------|---------|
| $x$ | Task input |
| $\varphi(x)$ | Feature matrix |
| $y(x)$ | Desired task output |
| $W$ | Weights of target function |
| $\hat{y}(x)$ | Output of model |
| $\theta$ | Weights of linear model |
| $\Lambda$ | Covariance matrix of $W$ |
| $\lambda_i$ | Element of $\Lambda$ |
| $n$ | # of training samples |
| $p$ | # of model parameters |
| $P$ | # of total features |
| $d$ | Task output dimensionality |
| $m$ | Task intrinsic dimensionality |

### 3.1 MODEL SETUP

We defer the full details of our theoretical model setup to App A. Here we present an overview: we consider a regression task with input $x \in \mathbb{R}^m$ and a feature matrix $\varphi(x) \in \mathbb{R}^{d \times P}$, where $m$ is input dimensionality, $d$ is output dimensionality and $P$ is the number of features. We assume that over the data distribution, the features are distributed i.i.d from a unit Gaussian: $\varphi(x)_{i,j} \sim \mathcal{N}(0, 1)$. We consider the limit when $P \to \infty$. Suppose our regression target function $y : \mathbb{R}^m \to \mathbb{R}^d$ is constructed linearly from $\varphi(x)$: $y(x) = \varphi(x)W$. Assume $\mathbb{E}[W] = 0$ and $\mathbb{E}[WW^T] = \Lambda$, where $\Lambda$ is diagonal and $\mathrm{Tr}(\Lambda)$ is finite. Suppose we aim to approximate $y(x)$ using a model $\hat{y}(x)$ constructed as follows: $\hat{y}(x) = \varphi(x) \begin{bmatrix} I \\ 0 \end{bmatrix} \theta$, where $\theta \in \mathbb{R}^{p \times 1}$ are model parameters and $p$ is the number of parameters. This corresponds to the model only being able to control $p$ of the $P$ true underlying parameters in the construction of $y$. We decompose $\Lambda$ blockwise as: $\Lambda = \begin{bmatrix} \Lambda_1 & 0 \\ 0 & \Lambda_2 \end{bmatrix}$ where $\Lambda_1 \in \mathbb{R}^{p \times p}$ and $\Lambda_2 \in \mathbb{R}^{(P-p) \times (P-p)}$. To capture the dependence of the target function on the input dimensionality $m$, we parameterize $\Lambda$ as having individual elements $\lambda_i = c \left[ i^{-\Omega^{-m}} - (i+1)^{-\Omega^{-m}} \right]$. This corresponds to the number of effective dimensions of variation of $W$ scaling exponentially with $m$, which is consistent with prior work (McRae et al., 2020). We also produce versions of our theoretical results without setting a specific form for $\lambda_i$. Finally, we consider learning $\theta$ as the minimum norm interpolating solution.

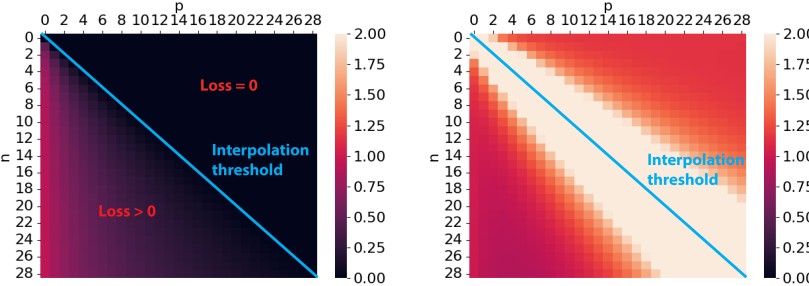

Figure 1: Expected training (left) and test (right) set error in a toy model of NN generalization as a function of the number of samples $n$ and the number of model parameters $p$. The output dimensionality is set as $d = 1$.

### 3.2 THEORETICAL PROPERTIES

Next, we theoretically analyze the expected training and test set error of the above model.

**Theorem 1.** *Given a target function $y$ and model $\hat{y}$ estimated as described above, in the limit that $P \to \infty$, the expected test loss when averaging over $x$ and $W$ is:*

$$\lim_{P \to \infty} \mathbb{E}\left[|y(x) - \hat{y}(x)|^2\right]$$

$$= d\operatorname{Tr}(\Lambda_2)F(dn, p) - d\frac{\min(dn, p)}{p}\operatorname{Tr}(\Lambda_1) + d\operatorname{Tr}(\Lambda)$$

$$= dF(dn, p)c(p+1)^{-\Omega^{-m}} - d\frac{\min(dn, p)}{p}(c - c(p+1)^{-\Omega^{-m}}) + dc \quad (1)$$

*The expected training loss is:*

$$\lim_{P \to \infty} \frac{1}{n}\mathbb{E}\left[\|y(X) - \hat{y}(X)\|_2^2\right] = \operatorname{Tr}(\Lambda_2)(dn - \min(dn, p)) = \frac{dn - \min(dn, p)}{n}c(p+1)^{-\Omega^{-m}} \quad (2)$$

*with $F(n, p)$ defined as $F(n, p) = \mathbb{E}\left[\|R^\dagger\|_F^2\right]$ where $R \in \mathbb{R}^{n \times p}$ has elements drawn i.i.d. from $\mathcal{N}(0, 1)$.*

Please see App B for a proof and App D for more details on $F(n, p)$. Under general $\lambda_i$, training and test error grow with $Tr(\Lambda_2)$; the specific rate at which they grow or shrink with parameters $m$ and $p$ depends on how rapidly $Tr(\Lambda_2)$ grows with $m$ and shrinks with $p$. Under the specific parameterization for $\lambda_i$ in Eqn 19, Fig 1 plots the value of the training and set error for varying $n$ and number of parameters, holding $d = 1$. Observe that there is a clear interpolation threshold at $p = n$ where the training loss becomes zero; for $p \geq n$ the model has sufficient capacity to perfectly interpolate the training set. The training loss is positive and increases with $n$ in the underparameterized regime ($p < n$) since the model lacks the capacity to fit increasing amounts of data $n$. At the $p = n$ threshold, the test loss dramatically increases, then decreases as $p$ increases beyond $n$. This is consistent with empirically observed behavior of overparameterized NNs (Belkin et al., 2019). Similarly, test loss decreases as $n$ increases beyond $p$.

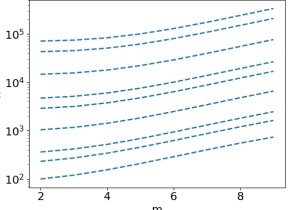

Figure 2: Theoretically predicted trend of $m$ vs. $n$ to achieve a test loss of $1.2$ on a sine wave regression task. Each line indicates a different fully connected NN with a different width and depth. $m$ increases approximately exponentially with $n$.

### 3.3 EMPIRICAL VALIDATION

We empirically validate our theoretical model on a parametrically-variable modular sine wave regression task with targets constructed as $y(x) = \frac{1}{\sqrt{k}}\sum_{i=1}^{k}\sum_{j=1}^{\tau} a_{ij}\sin(\omega_{ij}U_i^T x + \phi_{ij})$, where

$x \in \mathbb{R}^m$ are inputs, $y(x) \in \mathbb{R}$ are outputs, $a_{ij}, \omega_{ij}, \phi_{ij} \in \mathbb{R}$, $U_i \in \mathbb{R}^m$ are parameters chosen randomly for each target function and $k$ and $\tau$ are fixed (see App E for further task details). We train fully connected ReLU-activated NNs of varying depth and width on the task. The task allows us to quantify how NN generalization depends on a number of factors such as the dimensionality $m$ of the task input, the number of model parameters $p$, the number of samples $n$ and the number of modules $k$ in the construction of the target function. We defer the full details of the empirical validation to App A. In summary, our theoretical model matches empirical trends of neural network training and test error (see App E Fig 5). Nevertheless, we note two key discrepancies between empirical and predicted trends: first, the test loss is empirically larger than predicted for low training data. We hypothesize that this may be because of difficulty optimizing for small $n$: indeed, we find that the training loss is larger than expected for small $n$ (in the overparameterized regime ($n < p$), we expect a training loss of $0$). Second, the error spike at the interpolation threshold is smaller than theoretically predicted. This again may be due to incomplete optimization, given that the interpolation threshold spike can be viewed as highly adverse fitting to spurious training set patterns.

Our theoretical model predicts that the number of training samples required for generalization to a fixed error rate scales *exponentially* with task dimensionality $m$ (Fig 2). This raises a practical challenge for high-dimensional problems, which may require massive amounts of training data. Next, we aim to circumvent this exponential scaling.

## 4 USING MODULARITY TO GENERALIZE IN HIGH DIMENSIONS

So far, we have shown that a theoretical model treating NN learning as linear regression can closely model the generalization trends of actual NNs. Under our model, the number of training samples required to generalize on a task with $m$ dimensional input scales exponentially with $m$. Now, we demonstrate that *modular* NNs (in contrast to the *monolithic* NNs studied so far) can avoid this exponential dependence on $m$ *for tasks with an underlying modular structure*. We will consider modular networks in which model parameters are divided into separate modules, each of which processes a projection of the input; monolithic (or non-modular) networks in this context will correspond to networks without an explicit separation of parameters into modules. We first demonstrate the theoretical advantages of modular NNs under a specific form of modularity, then develop a modular NN learning rule to learn a task's underlying modular structure. We then empirically validate our approach and demonstrate that our approach can learn the true modules underlying the task.

### 4.1 SAMPLE COMPLEXITY OF MODULAR LEARNING

Recall in Sec 3, modeling a NN as a linear function of its parameters successfully captured its generalization properties. We aim to use this model to demonstrate the improved generalization of modular learning. For analytical traceability, we restrict our analysis to a *specific* modular setting that captures crucial aspects of many real-world modular learning scenarios. In practical settings, modules often handle low-dimensional inputs, such as attention maps in Andreas et al. (2016b). As such, we assume modules receive projected versions of the task input $x \in \mathbb{R}^m$. Our theoretical analysis will assume linear projections, but our method and experiments are also applied to *non-linear* projections. Moreover, we assume module outputs are summed to produce a final output, a feature of architectures such as Mixture of Experts (Shazeer et al., 2017). Specifically, consider a modular NN constructed as a linear combination of *general* NNs (each constituting a module) of low-dimensional projections of the input:

$$\hat{y}(x) = \frac{1}{\sqrt{K}} \sum_{j=1}^{K} \hat{y}_j(\hat{U}_j^T x), \tag{3}$$

where $\hat{U}_j \in \mathbb{R}^{m \times b}$ is a linear projection, $\hat{y}_j : \mathbb{R}^b \to \mathbb{R}^d$ is a NN. We will assume that $b$ is small, creating a bottleneck to each module's input. We normalize by $\frac{1}{\sqrt{K}}$ to make the scale of $\hat{y}(x)$ invariant to $K$. Note that each module $\hat{y}_j$ is itself a monolithic NN with *arbitary architecture*; we do not restrict the form of the modules themselves. Using the linearity assumption of Eqn 18 (namely, the model is a linear function of parameters), we may model this as:

$$\hat{y}_j(\hat{U}_j^T x) = \varphi^{(U)}(x) \mathbf{F}(\hat{U}_j) + \varphi^{(W)}(x) \begin{bmatrix} I \\ 0 \end{bmatrix} \theta_j, \tag{4}$$

where $\theta_j \in \mathbb{R}^{1 \times p}$ are the parameters of $\hat{y}_j$, $\varphi^{(U)}(x) \in \mathbb{R}^{d \times mb}$ and $\varphi^{(W)}(x) \in \mathbb{R}^{d \times P}$ are feature matrices, $\mathbf{F}(\cdot)$ denotes flattening a matrix into a vector, and we consider the limit when $P \to \infty$. As before, we assume the features are distributed i.i.d from a unit Gaussian: $\varphi^{(U)}(x)_i \sim \mathcal{N}(0, I), \varphi^{(W)}(x)_i \sim \mathcal{N}(0, I)$. We may then write $\hat{y}$ as a linear model:

$$\hat{y}(x) = \varphi^{(U)}(x) \frac{1}{\sqrt{K}} \sum_{j=1}^{K} \mathbf{F}(\hat{U}_j) + \varphi^{(W)}(x) \begin{bmatrix} I \\ 0 \end{bmatrix} \frac{1}{\sqrt{K}} \sum_{j=1}^{K} \theta_j. \tag{5}$$

Next, we assume that the regression target $y$ has the same modular structure with $k$ modules:

$$y(x) = \frac{1}{\sqrt{k}} \sum_{j=1}^{k} y_j(U_j^T x), \tag{6}$$

where $U_j \in \mathbb{R}^{m \times b}$ are the true projection directions and $y_j : \mathbb{R}^b \to \mathbb{R}^c$ are the true modules underlying the target. We apply the same linearity assumption to find:

$$y(x) = \varphi^{(U)}(x) \frac{1}{\sqrt{k}} \sum_{j=1}^{k} U_j + \varphi^{(W)}(x) \frac{1}{\sqrt{k}} \sum_{j=1}^{k} W_j \tag{7}$$

where $W_j \in \mathbb{R}^P$ are the parameters of $y_j$. As in the case of monolithic networks, we assume $\mathbb{E}[W_j] = 0$ and $\mathbb{E}[W_j W_j^T] = \Lambda$ where $\Lambda$ is diagonal. In this case, observe that each $W_j$ parameterizes a function with a $b$-dimensional input; thus it is appropriate to assume $W_j$ is distributed as if $m = b$:

$$\lambda_i = c \left[ i^{-\Omega^{-b}} - (i+1)^{-\Omega^{-b}} \right] \tag{8}$$

To preserve spherical symmetry in the distribution of $U_j$, we assume that $\mathbb{E}[U_j] = 0$ and $\mathbb{E}[\mathbf{F}(u_j)\mathbf{F}(u_j)^T] = I$. To complete our model definition, we define $\varphi(x) \in \mathbb{R}^{d \times (mb+P)}$ as the concatenation of $\varphi^{(U)}(x)$ and $\varphi^{(W)}(x)$: $\varphi(x) = \left[ \varphi^{(U)}(x), \varphi^{(W)}(x) \right]$. We may then write:

$$y(x) = \varphi(x) \begin{bmatrix} I \\ 0 \end{bmatrix} \overline{\theta} \tag{9}$$

where $\overline{\theta} \in \mathbb{R}^{mb+p}$ is defined as: $\overline{\theta} = \frac{1}{\sqrt{K}} \left[ \sum_{j=1}^{K} \mathbf{F}(\hat{U}_j), \sum_{j=1}^{K} \theta_j \right]$. Similarly, we may write: $\hat{y}(x) = \varphi(x)\overline{W}$, where $\overline{W} \in \mathbb{R}^{mb+P}$ is defined as: $\overline{W} = \frac{1}{\sqrt{k}} \left[ \sum_{j=1}^{k} \mathbf{F}(U_j), \sum_{j=1}^{k} W_j \right]$. Note that this model is nearly identical to that of monolithic NNs in Sec 3: they key difference is the different distribution of $\overline{W}$. $\overline{W}$ has covariance $\overline{\Lambda} = \begin{bmatrix} I & 0 \\ 0 & \Lambda \end{bmatrix}$ where $\Lambda$ is not dependent on $m$; $\bar{\Lambda}_i$ is defined analogously to $\Lambda_i$. Assuming $\hat{y}$ is trained to minimize squared loss on a training set, we may adapt Theorem 1 to this setting to compute the expected training and test loss of modular networks:

**Theorem 2.** *Given a target function $y$ and model $\hat{y}$ estimated as described above, in the limit that $P \to \infty$, the expected test loss when averaging over $x$ and $\overline{W}$ is:*

$$\lim_{P \to \infty} \mathbb{E} \left[ \|y(x) - \hat{y}(x)\|^2 \right]$$

$$= d \operatorname{Tr}(\overline{\Lambda}_2) F(dn, p) - d \frac{\min(dn, p)}{p} \operatorname{Tr}(\overline{\Lambda}_1) + d \operatorname{Tr}(\overline{\Lambda})$$

$$= dF(dn, p)c(p+1)^{-\Omega^{-b}} - d \frac{\min(dn, p)}{p} \left( mb + c - c(p+1)^{-\Omega^{-b}} \right) + dmb + dc \tag{10}$$

*The expected training loss is:*

$$\lim_{P \to \infty} \frac{1}{n} \mathbb{E} \left[ \|y(X) - \hat{y}(X)\|_2^2 \right] = \frac{dn - \min(dn, p)}{n} \operatorname{Tr}(\bar{\Lambda}_2) = \frac{dn - \min(dn, p)}{n} c(p+1)^{-\Omega^{-b}} \tag{11}$$

*with $F(n, p)$ defined as:*

$$F(n, p) = \mathbb{E} \left[ \left\| R^\dagger \right\|_F^2 \right] \tag{12}$$

*where $R \in \mathbb{R}^{n \times p}$ has elements drawn i.i.d. from $\mathcal{N}(0, 1)$.*

Please see App C for a proof. Under the condition that $Tr(\bar{\Lambda}_2)$ is independent of $m$, we have that unlike for the monolithic network, the training loss does not depend on $m$, and the dependence of the test loss on $m$ is linear. Moreover, in the underparameterized regime ($dn > p$), the test loss becomes $(dF(dn, p) + d) \operatorname{Tr}(\bar{\Lambda}_2)$ which *does not depend on* $m$, implying the $n$ required to reach a specific loss can be bounded by a function of only $p$ (assuming some value of $p$ can achieve the desired loss). Thus, the sample complexity of modular NNs is *independent* of the task dimensionality. Note that this dimension independence holds regardless of the parameterization of $\lambda_i$; the only condition is that the modules must have a dimension-independent input bottleneck (i.e. the covariance $\Lambda$ of module parameters must be independent of $m$). This result suggests that, unlike monolithic NNs, modular NNs can scale to high-dimensional, modular problems without requiring intractable amounts of data.

## 4.2 MODULAR LEARNING RULE

Inspired by the improved theoretical generalizability of modular NNs, and the finding that modular architectures trained with gradient descent on a task often cannot exploit these efficiencies (Csordás et al., 2021; Mittal et al., 2022), we develop a modular learning rule that practically exhibits this advantage. Importantly, we now relax the assumption that module input projections are linear.

We consider modular regression tasks with targets constructed as:

$$y(x) = \sum_{j=1}^{k} y_j(x; U_j) \tag{13}$$

where $y_j$ are functions that depend on a potentially *nonlinear* projection of the input $x$ as represented by $y_j$ depending on both module projections $U_j$ and inputs $x$. Observe that this generalizes the linear projections considered before (in Eqn 6). Suppose we aim to model the target function by approximating the $U_j$ with $\hat{U}_j$ and the $y_j$ with $\hat{y}_j$ (parameterized as a neural network). We propose a kernel-based rule to learn the *initializations* of $\hat{U}_j$ from training data; this allows us to efficiently learn the modules $\hat{y}_j$. Assume we are provided a set of training data $y(X) \in \mathbb{R}^{dn \times 1}$. Given the modular structure, we first aim to approximate the data as:

$$y(X) \approx \sum_{i=1}^{K} \varphi(X; \hat{U}_i)\theta_i, \tag{14}$$

where $X \in \mathbb{R}^{n \times m}$, and $\varphi$ is an arbitrary nonlinearity applied elementwise to the input data such that $\varphi(X; \hat{U}_i) \in \mathbb{R}^{dn \times p}$, $\theta_i \in \mathbb{R}^p$ and $K$ is the number of expected modules.

We expect that if $\hat{U}_i = U_i$ and $\varphi$ is sufficiently expressive, then $y(X)$ can be well approximated. Assuming $pK > dn$, observe that the minimum norm solution for $\theta_i$ can be computed as:

$$\begin{bmatrix} \theta_1 \\ \theta_2 \\ \vdots \\ \theta_K \end{bmatrix} = \begin{bmatrix} \varphi(X; \hat{U}_1) & \varphi(X; \hat{U}_2) & \cdots & \varphi(X; \hat{U}_K) \end{bmatrix}^{\dagger} y(X) \tag{15}$$

In general, such a solution exists for any choice of $\hat{U}_i$. However, we hypothesize that if the $\hat{U}_i$ is far from $U_i$, then the norm of the $\theta_i$ will be large: intuitively, interpolating the data along the "incorrect" projection directions $\hat{U}_i$ will be more difficult. Thus, we optimize $\hat{U}_i$ to minimize the squared norm of the $\theta_i$. Specifically, we minimize:

$$\sum_{i=1}^{K} \|\theta_i\|_2^2 = y(X)^T \mathbf{K}^{-1} y(X) \tag{16}$$

where $\mathbf{K} \in \mathbb{R}^{dn \times dn}$ is a kernel matrix applied to the data corresponding to the following kernel: $\mathbf{K}(x_1, x_2) = \sum_{i=1}^{K} \varphi(x_1^T; \hat{U}_i)\varphi(x_2^T; \hat{U}_i)^T = \kappa(x_1, x_2; \hat{U}_i)$, where $\kappa(x_1, x_2; \hat{U}_i)$ is a module-conditional kernel between $x_1$ and $x_2$. Experimentally, we tailor $\kappa$ to the modular structure of the problem we consider. Note that the above analysis only applies when $\phi$ is sufficiently expressive (i.e. $pK > dn$), which is a natural assumption for typically overparameterized models like neural

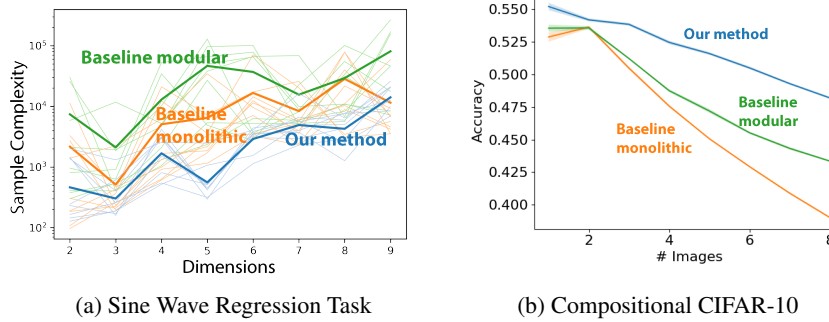

(a) Sine Wave Regression Task     (b) Compositional CIFAR-10

Figure 3: Comparison of our method with baselines of modular and monolithic architectures trained from random initialization on the sine wave regression task (a) and Compositional CIFAR-10 (b). (a): Required training sample size to achieve a desired rest error vs. # of input dimensions. Each light line indicates a different model architecture specified in App E averaged over five random seeds. The bold lines show averages over the light lines. (b): Accuracy vs. # of component images with a fixed number of training samples. Margins indicate standard errors over five random seeds.

networks. When models do not satisfy this assumption, Eqn 15 yields a solution minimizing the (generally non-zero) error between the predicted and true $y(X)$. Importantly, in this case, minimizing the norm of $\theta$ with respect to $\hat{U}_i$ *may not necessarily yield* a lower prediction error.

App E describes the specific choice of $\kappa$. Alg 1 shows the full procedure to find a single module projection $\hat{U}_i$; each step of the algorithm simply applies gradient on Eqn 16 with respect to $\hat{U}_i$. We repeat this procedure $K$ times with different random initializations to find the initial values of all $K$ module projections in our architecture. Then, we train all module parameters (including the $\hat{U}_i$) via gradient descent on the task loss. We stress that our approach is applicable to a fairly general set of modular architectures of the form $\sum_j \hat{y}_j(x; \hat{U}_j)$: *it does not restrict modules to receive only linear projections of inputs*, and, moreover, *does not restrict the form of the modules*.

### 4.3 EXPERIMENTAL RESULTS

We evaluate the generalizability of our method on a modular NN vs. baselines of a randomly initialized monolithic and modular NN trained on 1) sine wave regression tasks of varying dimensionality (fixing $k = m$), 2) a nonlinear variant of the sine wave regression task where the task has a nonlinear module structure, and 3) Compositional CIFAR-10 (based on Compositional MNIST (Jarvis et al., 2023)), a modular task in which each input consists of multiple CIFAR-10 images and the goal is to simultaneously predict the classes of all images; see App E Fig 7 for an illustration. In Compositional CIFAR-10, each input is constructed as a concatenation of $k$ flattened CIFAR-10 images (resulting in a $3072k$ dimensional vector) and target outputs are $k$-hot encoded $10k$ dimensional vectors encoding the class of each component image. The modular architectures are constructed as $\hat{y}(x) = \sum_{j=1}^{k} \hat{y}_j(x; U_j)$ where $y_j$ are fully connected, ReLU-activated networks, and the projection operation $U_j$ parameterizes the first layer. Monolithic architectures are normal fully connected ReLU-activated networks. See App E for further details on the datasets and the full experimental setup.

**Modular NNs empirically generalize better in and out-of-distribution** As shown in Fig 3, our modular method generalizes better compared with both the monolithic baseline method and the modular baseline method as evaluated by sample complexity for the sine wave regression task and accuracy for Compositional CIFAR-10. On both tasks, our method's advantage persists even on higher-dimensional inputs. Interestingly, on the sine wave task, the monolithic baseline outperforms the modular baseline, highlighting the difficulty of optimizing modular architectures. We also conduct experiments on additional variants of Compositional CIFAR-10 that test out-of-distribution generalization: 1) our method learns to classify *unseen* class combinations, thus generalizing combinatorially (App F Tab 3) and 2) our method is robust to small amounts of Gaussian noise added to training inputs (App F Tab 4), thus generalizing to small distribution shifts.

**Our learning rule finds the true task modules** Fig 4 computes a similarity score between our learned module projections ($\hat{U}$) and the target module projections ($U$) on the sine wave regression task; our method indeed aligns with the target modules. App F Fig 11 plots a low-dimensional representation of the target ($U$) and learned ($\hat{U}$) module projections: our learned NN initializations closely cluster around the target modules *without any task training*. On Compositional CIFAR-10, the learned module projections can be directly visualized as in App F Fig 12; here, our module initializations make each module sensitive to single component images *without any task training*, suggesting our approach promotes generalization by correctly learning the modular structure of a task. We also conduct ablation studies in App F Fig 9 which show that 1) our method performs nearly as well as using the ground-truth module directions and 2) allowing our learned module directions to adapt directly on the task loss improves performance.

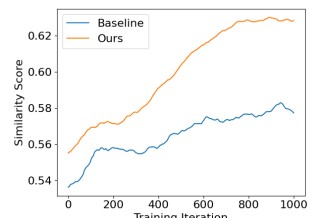

Figure 4: Average cosine similarity between learned and target module directions over training for a modular NN initialized with our method vs. random initialization (baseline).

**Our learning rule extends to nonlinear module projections** So far, we have considered modules ($y_j$ in Eqn 13) constructed as functions of $U_j^T x$, which is a linear function of both $U_j$ and $x$. Next, we consider a nonlinear variant of the sine wave regression task in which modules are a function of $||U_j - x||_2$, which is *nonlinear* in both $U_j$ and $x$. Similarly, the modular architecture is constructed nonlinearly as: $\frac{1}{\sqrt{K}} \sum_{j=1}^{K} \hat{y}_j(||\hat{U}_j - x||_2)$; see App E for further details. Tab 2 illustrates that our method significantly outperforms the baselines, indicating that our method extends to nonlinear settings as well.

Table 2: Comparison of our method with baselines on a nonlinear variant of the sine wave regression task where $k = m = 5$. Test loss values are evaluated with standard errors over 5 trials.

| Method | Test Loss |
|---|---|
| Baseline monolithic | $1.302 \pm 0.223$ |
| Baseline modular | $0.555 \pm 0.136$ |
| Our method | $0.393 \pm 0.099$ |

## 5 Discussion

Existing NN scaling laws show that in order to generalize on a task, monolithic NNs require an exponential number of training samples with the task's dimensionality. In this paper, we develop theory demonstrating that *modular NN* can break this scaling law: they only require only a *constant* number of samples to generalize in terms of task dimension when applied to modular tasks. To our knowledge, *we are the first to demonstrate such a result using explicit expressions for generalization error in modular NNs.* Based on this theoretical finding, we propose a novel learning rule for modular NNs and demonstrate its improved generalization, both in and out of distribution, on a sine wave regression task and Compositional CIFAR-10.

In pursuit of explicit, non-asymptotic expressions for generalization error in modular and monolithic NNs, we make strong theoretical assumptions consistent with previous literature, which we hope can be further relaxed in future work. Moreover, our results apply to a specific form of modularity that captures structures in common real-world modular architectures, but is not fully general. Notably, our theory considers linear module projections (although our method is also applied to non-linear projections), and both the theory and experiments assume the model output is the sum of module outputs. We expect that future analyses can demonstrate the benefits of modularity more widely. Finally, we find that while the theory predicts a task-dimension independent sample complexity for modular NNs, empirically we do not eliminate this dependence due to the difficulty of optimizing modular NNs in high dimensions. Nevertheless, our learning rule significantly eases this challenge.

Practically, we expect that modularity provides the most benefit for modular tasks with high-dimensional inputs; this is because the relative sample complexity improvement between non-modular and modular tasks is greater when task dimensionality increases. Indeed, as discussed in Sec 2, modularity empirically significantly enhances generalization in domains ranging from reinforcement learning and robotics to visual question answering and language modeling, all of which can be highly compositional and can have high-dimensional task inputs. We speculate that the modular structure of self-attention-based architectures may explain their success in many of these domains. Our findings provide a step toward fundamentally understanding how modularity can be better applied to solve high-dimensional generalization problems.

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

# A  THEORETICAL MODEL OF NEURAL NETWORK GENERALIZATION

## A.1  SETUP

We consider a regression task with input $x \in \mathbb{R}^m$ and a feature matrix $\varphi(x) \in \mathbb{R}^{d \times P}$ such that over the data distribution, the features are distributed i.i.d from a unit Gaussian: $\varphi(x)_{i,j} \sim \mathcal{N}(0, 1)$. We consider the limit when $P \to \infty$. Suppose our regression target function $y : \mathbb{R}^m \to \mathbb{R}^d$ is constructed linearly from $\varphi(x)$:

$$y(x) = \varphi(x)W, \tag{17}$$

where $W \in \mathbb{R}^{P \times 1}$. To accommodate multidimensional outputs, note that we shape input features $\phi(x)$ as a matrix (with one row for each output dimension) and parameters $W$ as a vector. This choice is more general than parameterizing each output dimension independently (this can be captured as a special case of our approach) and , moreover, aligns with prior theoretical literature (Jacot et al., 2018). Assume $\mathbb{E}[W] = 0$ and $\mathbb{E}[WW^T] = \Lambda$, where $\Lambda$ is diagonal and $\mathrm{Tr}(\Lambda)$ is finite. Suppose we aim to approximate $y(x)$ using a model $\hat{y}(x)$ constructed as follows:

$$\hat{y}(x) = \varphi(x) \begin{bmatrix} I \\ 0 \end{bmatrix} \theta, \tag{18}$$

where $\theta \in \mathbb{R}^{p \times 1}$ are model parameters and $p$ is the number of parameters. This corresponds to the model only being able to control $p$ of the $P$ true underlying parameters in the construction of $y$. We decompose $\Lambda$ blockwise as: $\Lambda = \begin{bmatrix} \Lambda_1 & 0 \\ 0 & \Lambda_2 \end{bmatrix}$ where $\Lambda_1 \in \mathbb{R}^{p \times p}$ and $\Lambda_2 \in \mathbb{R}^{(P-p) \times (P-p)}$.

We make a specific choice of parameterization for the individual elements $\lambda_i$ of $\Lambda$:

$$\lambda_i = c \left[ i^{-\Omega^{-m}} - (i+1)^{-\Omega^{-m}} \right] \tag{19}$$

for some constants $c$ and $\Omega$. We justify this choice as follows: we define the *effective dimensionality* of $\Lambda$ as $\frac{(\sum_i \lambda_i)^2}{\sum_i \lambda_i^2}$ (this measure approximates the $\ell_0$ norm (Krishnan et al., 2011), and thus can be used as a measure of $\Lambda$'s dimensionality). For large $m$, this can be approximated as: $\frac{(\sum_i \lambda_i)^2}{\sum_i \lambda_i^2} \approx \Omega^{2m}$; we interpret this as $y$ having $\Omega^{2m}$ free parameters. This is consistent with the observation that regression on an $m$-dimensional input space has a function space that scales *exponentially* with $m$ (McRae et al., 2020). Intuitively, this is because the function must have enough free parameters to express values at all points in volume of its input space, and volume scales exponentially with $m$ (Sharma & Kaplan, 2022).

Next, suppose we are given a set of training data $X$ with associated feature matrix transformation $\varphi(X) \in \mathbb{R}^{dn \times P}$, where $n$ is the number of data points. Suppose $\theta$ is optimized to find the minimum norm interpolating solution to the data:

$$\min_\theta \left\{ \left\| \varphi(X) \begin{bmatrix} I \\ 0 \end{bmatrix} \theta - \varphi(X)W \right\|_2^2 + \frac{\gamma}{2} \|\theta\|_2^2 \right\}, \tag{20}$$

where $\gamma \to 0$. Recall that the solution can be found as:

$$\theta = \left[ \varphi(X) \begin{bmatrix} I \\ 0 \end{bmatrix} \right]^\dagger \varphi(X)W. \tag{21}$$

Thus, the model's prediction on a point $x$ is:

$$\hat{y}(x) = \varphi(x) \begin{bmatrix} I \\ 0 \end{bmatrix} \left[ \varphi(X) \begin{bmatrix} I \\ 0 \end{bmatrix} \right]^\dagger \varphi(X)W. \tag{22}$$

## A.2  EMPIRICAL VALIDATION

Next, we empirically validate our theoretical model on a parametrically-variable modular sine wave regression task (see App E for task details). The task allows us to quantify how NN generalization depends on the number of dimensions of variation such as the dimensionality $m$ of the task input,

the number of model parameters $p$, the number of samples $n$ and the number of modules $k$ in the construction of the target function. Note that our theoretical model does not include the number of modules $k$ since it does not explicitly construct the target modularly. Thus, directly applying our theory, we would expect the loss to be invariant to $k$. Intuitively, this is because varying $k$ increases the complexity of the task in a way that is irrelevant for generalization.

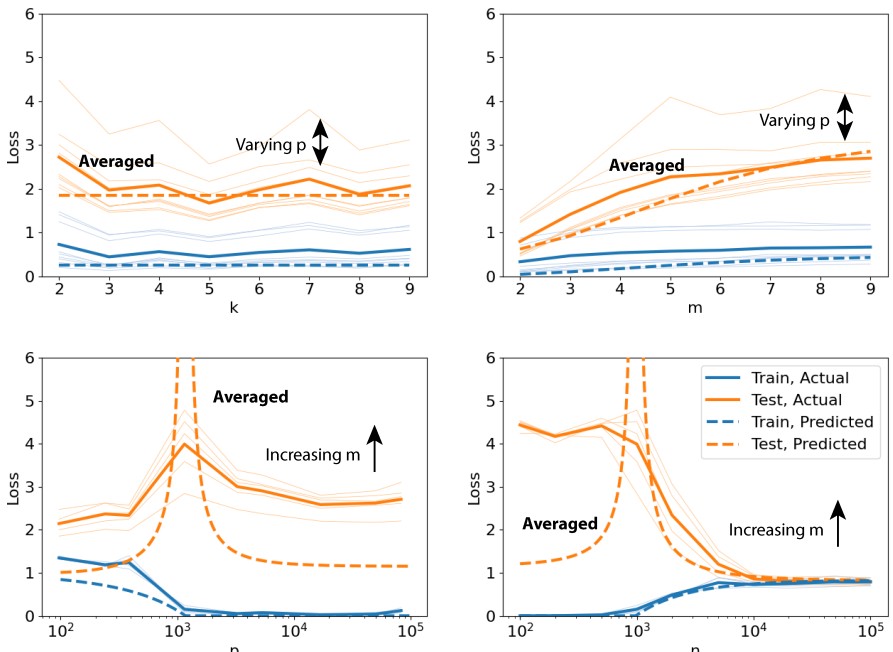

Figure 5: Empirical trends of training (blue) and test (orange) loss over four parametric variations for a NN trained on a sine wave regression task. The parameters varied are: $k$ (number of modules), $m$ (input dimensionality), $p$ (model size) and $n$ (training set size). In the first two plots, each line indicates a different model architecture, and in the last two plots, each line indicates a different choice of $m$ between $5$ and $9$, with $n/p$ fixed at $1000/1153$ respectively (left/right). The light lines are averaged over all other parameters, and bold lines show averages over the light lines. Dashed lines show theoretical predictions.

**Trends of generalization error of NNs** Now, we train NNs of various architectures on the task and observe error trends as a function of $k, m, p$ and $n$. We fit parameters $c$ and $\Omega$ of our theoretical model to our task. Furthermore, because each parameter in our NNs may not correspond to a single parameter in our theoretical model, we use a linear scaling of the number of true NN parameters to estimate the number of effective parameters in our theoretical model; specifically, we estimate $p = \alpha p'$ where $p'$ is the actual number of NN parameters. See App E for more details.

Fig 5 shows that our theoretical model can capture many trends of the training and test loss as a function of $k, m, p$ and $n$. In particular, our model predicts the invariance of loss to $k$, the sub-linear increase in loss with $m$, and the double descent behavior of loss with $p$ and $n$. Notably, our model predicts the empirical location of the interpolation threshold as seen in the last two plots of Fig 5.

We note two key discrepancies between our theory and empirical results: first, the loss is empirically larger than predicted for small amounts of training data, and second, the error spike at the interpolation threshold is smaller than predicted by the theory.

We believe the first discrepancy is due to imperfect optimization of neural networks, especially in low data regimes. Note that the linearized analysis assumes that the linear model solution finds the exact global optimum. However, the actual optimization landscape for modular architectures is highly non-convex, and the global optimum may not be found especially for small datasets (indeed, we find a significant discrepancy between predicted and actual training loss values for small data size $n$; in

the overparameterized regime, the predicted training error is exactly 0). We believe this causes the discrepancy between predicted and actual test errors in low data regimes.

We hypothesize that the second discrepancy is also partly due to imperfect optimization. This is because the interpolation threshold spike can be viewed as highly adverse fitting to spurious training set patterns. This imperfect optimization is more pronounced at smaller $m$. Despite these discrepancies, we nevertheless find that our theory precisely captures the key trends of empirical test error.

Finally, we consider the trend between $m$ and $n$ implied by our model. As Fig 2 reveals, for various non-modular architectures, the sample complexity grows approximately exponentially with the task dimensionality, consistent with prior theoretical observations (see Sec 2.2). This implies that generalizing on high-dimensional problems can require a massive number of samples.

## B  PROOF OF THEOREM 1

*Proof.* **Test set error** We first compute the expected test set error. Note that the squared error can be written as:

$$||\hat{y}(x) - y(x)||^2 = \left\| \varphi(x) \begin{bmatrix} I \\ 0 \end{bmatrix} \left[ \varphi(X) \begin{bmatrix} I \\ 0 \end{bmatrix} \right]^\dagger \varphi(X)W - \varphi(x)W \right\|^2 = \left\| \varphi(x) \left[ \begin{bmatrix} I \\ 0 \end{bmatrix} \left[ \varphi(X) \begin{bmatrix} I \\ 0 \end{bmatrix} \right]^\dagger \varphi(X) - I \right] W \right\|^2 \tag{23}$$

For notational convenience, define $A$ as the first $p$ columns of $\varphi(X)$ and $B$ as the remaining $P - p$ columns such that $\varphi(X) = [A, B]$. Also, define $M = \begin{bmatrix} I \\ 0 \end{bmatrix} \left[ \varphi(X) \begin{bmatrix} I \\ 0 \end{bmatrix} \right]^\dagger \varphi(X) = \begin{bmatrix} I \\ 0 \end{bmatrix} A^\dagger [A, B] = \begin{bmatrix} A^\dagger A & A^\dagger B \\ 0 & 0 \end{bmatrix}$. Then, using the cyclic property of trace, the squared error can be written as

$$||\varphi(x)(M-I)W||^2 = \mathrm{Tr}\left(\varphi(x)(M-I)WW^T(M-I)^T\varphi(x)^T\right) = \mathrm{Tr}\left((M-I)WW^T(M-I)^T\varphi(x)^T\varphi(x)\right) \tag{24}$$

Next, we can take the expectation with to $x$ and $W$ and use the fact that $\mathbb{E}[\varphi(x)^T\varphi(x)] = dI$ and $\mathbb{E}[WW^T] = \Lambda$ to find that:

$$\mathbb{E}\left[||\hat{y}(x) - y(x)||^2\right] = d\mathbb{E}\left[\mathrm{Tr}((M-I)\Lambda(M-I)^T)\right] \tag{25}$$

Finally, expanding:

$$d\mathbb{E}\left[\mathrm{Tr}((M-I)\Lambda(M-I)^T)\right] = d\mathbb{E}\left[\mathrm{Tr}(M^TM\Lambda)\right] - 2d\mathbb{E}\left[\mathrm{Tr}(M\Lambda)\right] + d\,\mathrm{Tr}(\Lambda) \tag{26}$$

Next, we compute $\mathbb{E}\left[\mathrm{Tr}(M^TM\Lambda)\right]$:

$$\mathbb{E}\left[\mathrm{Tr}(M^TM\Lambda)\right] = \mathbb{E}\left[\mathrm{Tr}\left(\begin{bmatrix} A^\dagger A & 0 \\ B^T A^{\dagger T} & 0 \end{bmatrix}\begin{bmatrix} A^\dagger A & A^\dagger B \\ 0 & 0 \end{bmatrix}\Lambda\right)\right] = \mathbb{E}\left[\mathrm{Tr}\left(\begin{bmatrix} A^\dagger A & A^\dagger AA^\dagger B \\ B^T A^{\dagger T}A^\dagger A & B^T A^{\dagger T}A^\dagger B \end{bmatrix}\Lambda\right)\right] \tag{27}$$

Next, we decompose $\Lambda$ blockwise as:

$$\Lambda = \begin{bmatrix} \Lambda_1 & 0 \\ 0 & \Lambda_2 \end{bmatrix} \tag{28}$$

where $\Lambda_1 \in \mathbb{R}^{p \times p}$ and $\Lambda_2 \in \mathbb{R}^{(P-p) \times (P-p)}$. Then:

$$\mathbb{E}\left[\mathrm{Tr}(M^TM\Lambda)\right] = \mathbb{E}\left[\mathrm{Tr}\left(\begin{bmatrix} A^\dagger A\Lambda_1 & A^\dagger AA^\dagger B\Lambda_2 \\ B^T A^{\dagger T}A^\dagger A\Lambda_1 & B^T A^{\dagger T}A^\dagger B\Lambda_2 \end{bmatrix}\right)\right] = \mathbb{E}\left[\mathrm{Tr}(A^\dagger A\Lambda_1)\right] + \mathbb{E}\left[\mathrm{Tr}(A^{\dagger T}A^\dagger B\Lambda_2 B^T)\right] \tag{29}$$

Next, consider, $\mathbb{E}[\mathrm{Tr}(M\Lambda)]$:

$$\mathbb{E}[\mathrm{Tr}(M\Lambda)] = \mathbb{E}\left[\mathrm{Tr}\left(\begin{bmatrix} A^\dagger A & A^\dagger B \\ 0 & 0 \end{bmatrix}\Lambda\right)\right] = \mathbb{E}\left[\mathrm{Tr}(A^\dagger A\Lambda_1)\right] \tag{30}$$

Combining this result with the earlier result, the expected squared error can be expressed as:

$$\mathbb{E}\left[||\hat{y}(x) - y(x)||^2\right] = d\mathbb{E}\left[\mathrm{Tr}(A^{\dagger T}A^\dagger B\Lambda_2 B^T)\right] - d\mathbb{E}\left[\mathrm{Tr}(A^\dagger A\Lambda_1)\right] + d\,\mathrm{Tr}(\Lambda) \tag{31}$$

Next, we evaluate $\mathbb{E}\left[\text{Tr}(A^{\dagger T} A^\dagger B \Lambda_2 B^T)\right]$. By linearity of trace, and the independence of $A$ and $B$, we have:

$$\mathbb{E}\left[\text{Tr}(A^{\dagger T} A^\dagger B \Lambda_2 B^T)\right] = \text{Tr}\left(\mathbb{E}[A^{\dagger T} A^\dagger] \cdot \mathbb{E}[B \Lambda_2 B^T]\right) \tag{32}$$

Define the following quantities:

$$\alpha = \mathbb{E}\left[A_{:,1}^{\dagger T} A_{:,1}^\dagger\right] \tag{33}$$

and

$$\beta = \mathbb{E}\left[B_{1,:} \Lambda_2 B_{1,:}^T\right] \tag{34}$$

where $\cdot_{i,j}$ indicates the $i$th row and $j$th column of the argument. Note that by symmetry over the data points and output dimensions, both $\mathbb{E}\left[A^{\dagger T} A^\dagger\right] \in \mathbb{R}^{dn \times dn}$ and $\mathbb{E}\left[B \Lambda_2 B^T\right] \in \mathbb{R}^{dn \times dn}$ must be proportional to the identity matrix. Thus, the expectation of their top left entry is the same as the expectation of any other entry:

$$\mathbb{E}\left[A^{\dagger T} A^\dagger\right] = \alpha I \tag{35}$$

and

$$\mathbb{E}\left[B \Lambda_2 B^T\right] = \beta I \tag{36}$$

Then,

$$\mathbb{E}\left[\text{Tr}(A^{\dagger T} A^\dagger B \Lambda_2 B^T)\right] = \text{Tr}(\alpha I \beta I) = \alpha \beta n \tag{37}$$

To evaluate $\alpha$, observe that since $A$ has elements distributed from $\mathcal{N}(0, 1)$:

$$F(dn, p) = \mathbb{E}\left[\|A^\dagger\|_F^2\right] = \text{Tr}\left(\mathbb{E}[A^{\dagger T} A^\dagger]\right) \tag{38}$$

Using the definition of $\alpha$:

$$F(dn, p) = \alpha dn \tag{39}$$

Therefore, $\alpha = \frac{F(dn, p)}{dn}$. To evaluate $\beta$, note that $B_{1,:}$ has elements distributed from $\mathcal{N}(0, 1)$. Thus, $\beta$ is simply:

$$\beta = \mathbb{E}\left[B_{1,:} \Lambda_2 B_{1,:}^T\right] = \sum_{i=1}^{P-p} \mathbb{E}[B_{1,i}^2] \Lambda_{2,i} = \text{Tr}(\Lambda_2) \tag{40}$$

Substituting $\alpha$ and $\beta$ into the expression for $\mathbb{E}\left[\text{Tr}(A^{\dagger T} A^\dagger B \Lambda_2 B^T)\right]$:

$$\mathbb{E}\left[\text{Tr}(A^{\dagger T} A^\dagger B \Lambda_2 B^T)\right] = \text{Tr}(\Lambda_2) F(dn, p) \tag{41}$$

Next, we evaluate $\mathbb{E}\left[\text{Tr}(A^\dagger A \Lambda_1)\right]$. First, define the singular value decomposition of $A$ as $A = U \Sigma V^T$. Then, expanding $A$ and using the cyclic property of trace:

$$\mathbb{E}\left[\text{Tr}(A^\dagger A \Lambda_1)\right] = \mathbb{E}\left[\text{Tr}(V \Sigma^\dagger U^T U \Sigma V^T \Lambda_1)\right] = \mathbb{E}\left[\text{Tr}(V \Sigma^\dagger \Sigma V^T \Lambda_1)\right] \tag{42}$$

Applying the linearity of trace:

$$\mathbb{E}\left[\text{Tr}(A^\dagger A \Lambda_1)\right] = \text{Tr}\left(\mathbb{E}[V \Sigma^\dagger \Sigma V^T] \Lambda_1\right) \tag{43}$$

Now, we examine $\mathbb{E}\left[V \Sigma^\dagger \Sigma V^T\right] \in \mathbb{R}^{p \times p}$. First, note that $\Sigma^\dagger \Sigma$ is a diagonal matrix with entries 1 and 0: specifically, it has $\min(dn, p)$ 1s and remaining entries (if any) 0. Thus, we may write:

$$\mathbb{E}\left[V \Sigma^\dagger \Sigma V^T\right] = \sum_{i=1}^{\min(dn, p)} \mathbb{E}\left[V_{:,i} V_{:,i}^T\right] \tag{44}$$

Next, note that the distribution of $A$ is symmetric to rotations of its $p$ columns. Thus $V_{:,i}$ must also have a rotationally symmetric distribution. Since $\|V_{:,i}\|_2 = 1$, $\mathbb{E}\left[V_{:,i} V_{:,i}^T\right] = \frac{1}{p} I$. Thus:

$$\mathbb{E}\left[V \Sigma^\dagger \Sigma V^T\right] = \frac{\min(dn, p)}{p} I \tag{45}$$

Substituting into the expression for $\mathbb{E}\left[\text{Tr}(A^\dagger A \Lambda_1)\right]$:

$$\mathbb{E}\left[\text{Tr}(A^\dagger A \Lambda_1)\right] = \frac{\min(dn, p)}{p} \text{Tr}(\Lambda_1) \tag{46}$$

Combining the results from earlier, the expected squared error can be written as:

$$\mathbb{E}\left[||\hat{y}(x) - y(x)||^2\right] = d\,\mathrm{Tr}(\Lambda_2)F(dn, p) - d\frac{\min(dn, p)}{p}\,\mathrm{Tr}(\Lambda_1) + d\,\mathrm{Tr}(\Lambda) \tag{47}$$

Next, using the definition of $\lambda_i = c\left[i^{-\Omega^{-m}} - (i+1)^{-\Omega^{-m}}\right]$, observe that:

$$\mathrm{Tr}(\Lambda_1) = \sum_{i=1}^{p} \lambda_i = \sum_{i=1}^{p} c\left[i^{-\Omega^{-m}} - (i+1)^{-\Omega^{-m}}\right] = c - c(p+1)^{-\Omega^{-m}} \tag{48}$$

$$\mathrm{Tr}(\Lambda_2) = \sum_{i=p+1}^{\infty} \lambda_i = \sum_{i=p+1}^{\infty} c\left[i^{-\Omega^{-m}} - (i+1)^{-\Omega^{-m}}\right] = c(p+1)^{-\Omega^{-m}} \tag{49}$$

$$\mathrm{Tr}(\Lambda) = \mathrm{Tr}(\Lambda_1) + \mathrm{Tr}(\Lambda_2) = c \tag{50}$$

We finally use the expressions for $\Lambda_1$ and $\Lambda_2$ to write the result in terms of $c$ and $\Omega$:

$$\mathbb{E}\left[||\hat{y}(x) - y(x)||^2\right] = dF(dn, p)c(p+1)^{-\Omega^{-m}} - d\frac{\min(dn, p)}{p}\left(c - c(p+1)^{-\Omega^{-m}}\right) + dc \tag{51}$$

**Training set error** Now, we compute the training set error. Writing out the summed training set error over all data points:

$$\|\hat{y}(X) - y(X)\|_2^2 = \left\|\varphi(X)\begin{bmatrix}I\\0\end{bmatrix}\left[\varphi(X)\begin{bmatrix}I\\0\end{bmatrix}\right]^\dagger \varphi(X)W - \varphi(X)W\right\|_2^2 = \left\|(AA^\dagger - I)[A, B]W\right\|_2^2$$

$$= \left\|[(AA^\dagger - I)A, (AA^\dagger - I)B]W\right\|_2^2 = \left\|[0, (AA^\dagger - I)B]W\right\|_2^2 \tag{52}$$

Expressing this squared norm as a trace and using the cyclic property of trace:

$$\left\|[0, (AA^\dagger - I)B]W\right\|_2^2 = \mathrm{Tr}\left(W^T[0, (AA^\dagger - I)B]^T[0, (AA^\dagger - I)B]W\right)$$

$$= \mathrm{Tr}\left([0, (AA^\dagger - I)B]^T[0, (AA^\dagger - I)B]WW^T\right) \tag{53}$$

Taking the expectation with respect to $W$:

$$\mathbb{E}\left[\mathrm{Tr}\left([0, (AA^\dagger - I)B]^T[0, (AA^\dagger - I)B]WW^T\right)\right] = \mathrm{Tr}\left(\mathbb{E}\left[B^T(AA^\dagger - I)^2B\right]\Lambda_2\right) \tag{54}$$

Note that $(AA^\dagger - I)^2 = I - AA^\dagger$ Again using the cyclic property of trace and the independence of $A$ and $B$, we find:

$$\mathrm{Tr}\left(\mathbb{E}[B^T(AA^\dagger - I)^2B]\Lambda_2\right) = \mathrm{Tr}\left(\mathbb{E}[I - AA^\dagger]\mathbb{E}[B\Lambda_2B^T]\right) \tag{55}$$

From the calculations for test set error, we have:

$$\mathbb{E}\left[B\Lambda_2B^T\right] = \mathrm{Tr}(\Lambda_2)I \tag{56}$$

Substituting into the earlier expression:

$$\mathrm{Tr}\left(\mathbb{E}[I - AA^\dagger]\mathbb{E}[B\Lambda_2B^T]\right) = \mathrm{Tr}\left(\mathbb{E}[I - AA^\dagger]\,\mathrm{Tr}(\Lambda_2)I\right) = \mathrm{Tr}(\Lambda_2)\mathbb{E}[\mathrm{Tr}(I - AA^\dagger)] \tag{57}$$

To evaluate $\mathbb{E}[\mathrm{Tr}(I - AA^\dagger)]$, we use the singular value decomposition of $A = U\Sigma V^T$ and the cyclic property of trace:

$$\mathbb{E}\left[\mathrm{Tr}(I - AA^\dagger)\right] = \mathbb{E}\left[\mathrm{Tr}(I - U\Sigma V^T V\Sigma^\dagger U^T)\right] = \mathbb{E}\left[\mathrm{Tr}(I - U\Sigma\Sigma^\dagger U^T)\right] = \mathbb{E}\left[\mathrm{Tr}(I - \Sigma\Sigma^\dagger)\right] \tag{58}$$

Observe that $A$ has full rank with probability 1. Thus, $\Sigma$ also has full rank with probability 1, implying that $\Sigma\Sigma^\dagger$ is with probability 1 a diagonal matrix with $\min(dn, p)$ 1s and remaining entries (if any) 0:

$$\mathbb{E}\left[\mathrm{Tr}(I - \Sigma\Sigma^\dagger)\right] = dn - \min(dn, p) \tag{59}$$

Substituting into the earlier expression, we have a result for the total training set error over all training points:

$$\mathbb{E}\left[\|\hat{y}(X) - y(X)\|_2^2\right] = \mathrm{Tr}(\Lambda_2)(dn - \min(dn, p)) \tag{60}$$

To arrive at the final result for expected training set error we simply divide by $n$ and express $\mathrm{Tr}(\Lambda_2)$ in terms of $c$ and $\Omega$

$$\frac{1}{n}\mathbb{E}[\|\hat{y}(X) - y(X)\|_2^2] = \frac{dn - \min(dn, p)}{n}\,\mathrm{Tr}(\Lambda_2) = \frac{dn - \min(dn, p)}{n}c(p+1)^{-\Omega^{-m}} \tag{61}$$

$\square$

## C  PROOF OF THEOREM 2

*Proof.* **Test set error** Using the same techniques as in the proof of Theorem 1, we may write the expected test set error in terms of $\Lambda$ as:

$$\mathbb{E}\left[||\hat{y}(x) - y(x)||^2\right] = d\operatorname{Tr}(\overline{\Lambda}_2)F(dn, p) - d\frac{\min(dn, p)}{p}\operatorname{Tr}(\overline{\Lambda}_1) + d\operatorname{Tr}(\overline{\Lambda}) \tag{62}$$

Using the definition $\lambda_i = c\left[i^{-\Omega^{-b}} - (i+1)^{-\Omega^{-b}}\right]$ and the assumption $p > mb$, observe that:

$$\operatorname{Tr}(\overline{\Lambda}_1) = mb + \sum_{i=1}^{p}\lambda_i = mb + \sum_{i=1}^{p}c\left[i^{-\Omega^{-b}} - (i+1)^{-\Omega^{-b}}\right] = mb + c - c(p+1)^{-\Omega^{-b}} \tag{63}$$

$$\operatorname{Tr}(\overline{\Lambda}_2) = \sum_{i=p+1}^{\infty}\lambda_i = \sum_{i=p+1}^{\infty}c\left[i^{-\Omega^{-b}} - (i+1)^{-\Omega^{-b}}\right] = c(p+1)^{-\Omega^{-b}} \tag{64}$$

$$\operatorname{Tr}(\overline{\Lambda}) = \operatorname{Tr}(\overline{\Lambda}_1) + \operatorname{Tr}(\overline{\Lambda}_2) = mb + c \tag{65}$$

Substituting these expressions, the expected test set error is:

$$\mathbb{E}\left[||\hat{y}(x) - y(x)||^2\right] = dF(dn, p)c(p+1)^{-\Omega^{-b}} - d\frac{\min(dn, p)}{p}\left(mb + c - c(p+1)^{-\Omega^{-b}}\right) + dmb + dc \tag{66}$$

**Training set error** Again, using the techniques in the proof of Theorem 1, we write the expected training set error in terms of $\bar{\Lambda}_2$

$$\frac{1}{n}\mathbb{E}\left[\|\hat{y}(X) - y(X)\|_2^2\right] = \frac{dn - \min(dn, p)}{n}\operatorname{Tr}(\bar{\Lambda}_2) \tag{67}$$

Using the expression for $\operatorname{Tr}(\bar{\Lambda}_2)$:

$$\frac{1}{n}\mathbb{E}\left[\|\hat{y}(X) - y(X)\|_2^2\right] = \frac{dn - \min(dn, p)}{n}c(p+1)^{-\Omega^{-b}} \tag{68}$$

$\square$

## D  PROPERTIES OF $F(n, p)$

In this section, we summarize some known properties about the function $F(n, p)$, which appears in Theorem 1. Recall that

$$F(n, p) = \mathbb{E}\left[\|R^\dagger\|_F^2\right],$$

where $R \in \mathbb{R}^{n \times p}$ has elements drawn i.i.d. from $\mathcal{N}(0, 1)$.

In the regime $|n - p| \geq 2$, an exact closed form is given by

$$F(n, p) = \frac{\min(n, p)}{|n - p| - 1} \qquad \text{if } |n - p| \geq 2.$$

Computing the square of the Frobenius norm of $R^\dagger$ is equivalent to finding $\operatorname{Tr}(R^\dagger R^{\dagger T}) = \operatorname{Tr}(R^{\dagger T} R^\dagger)$.

When $n - p \geq 2$, $R^\dagger R^{\dagger T}$ is a $p \times p$ matrix with Inverse-Wishart distribution of identity covariance, which has mean $\frac{1}{n-p-1}I$ (Von Rosen, 1988). Therefore, the expected value of its trace is $\frac{n}{n-p-1}$. Analogously, when $p - n \geq 2$, $R^{\dagger T} R^\dagger$ is a $n \times n$ matrix with Inverse-Wishart distribution of identity covariance, which has mean $\frac{1}{p-n-1}I$, so the expected value of its trace is $\frac{n}{p-n-1}$.

In the case where $p = n$, bound on the Frobenius norm of $R^\dagger$ are known (Szarek, 1991). However, for the cases where $|p - n| \leq 1$, $F(n, p)$ has no known explicit form, so it was computed by averaging over 100 random trials.

# E    EXPERIMENTAL DETAILS

## E.1    DATASET DETAILS

**Sine Wave Regression Task**    We construct a regression problem where the regression target is constructed as a sum of $k$ functions of *one-dimensional linear projections* of the input $x \in \mathbb{R}^m$. Specifically, the regression target function $y : \mathbb{R}^m \to \mathbb{R}$ is constructed as follows:

$$y(x) = \frac{1}{\sqrt{k}} \sum_{i=1}^{k} \sum_{j=1}^{\tau} a_{ij} \sin(\omega_{ij} U_i^T x + \phi_{ij}) \tag{69}$$

where $a_{ij}, \omega_{ij}, \phi_{ij} \in \mathbb{R}$, $U_i \in \mathbb{R}^m$, $\tau$ is the number of sine waves that comprise each function of one-dimensional linear projection $U_i^T x$. We sample these parameters of the target function independently from the following distributions: $a_{ij} \sim \mathcal{N}(0, 1)$, $\omega_{ij} \sim \mathcal{N}(0, 4\pi^2)$, $\phi_{ij} \sim \mathcal{U}(-\pi, \pi)$ where $\mathcal{N}$ denotes a Gaussian distribution and $\mathcal{U}$ denotes a uniform distribution. $u_i$ is drawn uniformly from a unit sphere. Note that the full function is made of $k$ separate functions of one-dimensional projections of $x$, each of which has $\tau$ sine components. We normalize by $\frac{1}{\sqrt{k}}$ so that $y(x)$ does not scale with $k$. Note that the task is *modular*: although the task takes an $m$ dimensional input, the target is constructed as a combination of $k$ *modules* operating on 1 dimensional projections of the input. Given the projections, parameterizing functions of the projections are sufficient to parameterize the full task.

A training dataset is generated by first drawing $n$ training samples $x$ from a mean-zero Gaussian: $x \sim N(0, I)$. Then, for each $x$, the regression target $y(x)$ is computed. The test dataset is constructed analogously. See Fig 6 for an illustrated example target function in one dimension.

Recall that any square integrable function can be approximated on a finite interval to arbitrary precision by a sufficiently large Fourier series. Thus, we may expect that as $T$ approaches infinity, the functional form in Equation 69 can express any function constructed as a sum of square integrable functions of the $U_i^T x$: $\sum_{i=1}^{k} y_i(U_i^T x)$ for any square integrable $y_i$.

Note that $m$ controls the dimensionality of $x$; thus, we may make the task test generalization over arbitrarily high dimensions by simply increasing $m$. This is significant because prior work shows that the number of samples required to generalize to a fixed precision on a regression problem scales *exponentially* with the intrinsic dimensionality of the task input (McRae et al., 2020; Sharma & Kaplan, 2022). Thus, even with a relatively simple task construction, we may expect to produce tasks with *arbitrary* difficulty as measured by sample complexity.

**Nonlinear Sine Wave Regression Task**    We also test our approach on a non-linear variation of our sine wave regression task. Recall that in the original sine wave regression task, outputs are constructed as:

$$y(x) = \frac{1}{\sqrt{k}} \sum_{i=1}^{k} \sum_{j=1}^{\tau} a_{ij} \sin(\omega_{ij} U_i^T x + \phi_{ij}) \tag{70}$$

where $U_i$ are module projection directions. Note that this task has linear module input projections (the projections $U_i^T x$ are linear functions of $x$ and $U_i$). In our nonlinear variant, we consider the following outputs constructed with non-linear module input projections:

$$y(x) = \frac{1}{\sqrt{k}} \sum_{i=1}^{k} \sum_{j=1}^{\tau} a_{ij} \sin(\omega_{ij} \|U_i - x\|_2 + \phi_{ij}) \tag{71}$$

where $U_i^T x$ is replaced with $\|U_i - x\|_2$, which is non-linear in both $U_i$ and $x$. Remaining task parameters are set the same way as in the original sine wave regression task.

**Compositional CIFAR-10**    We conduct experiments on a Compositional CIFAR-10 dataset inspired by the Compositional MNIST dataset of (Jarvis et al., 2023). In the task, combinations of $k$ CIFAR-10 images are concatenated together and the model is asked to predict the class of all component images

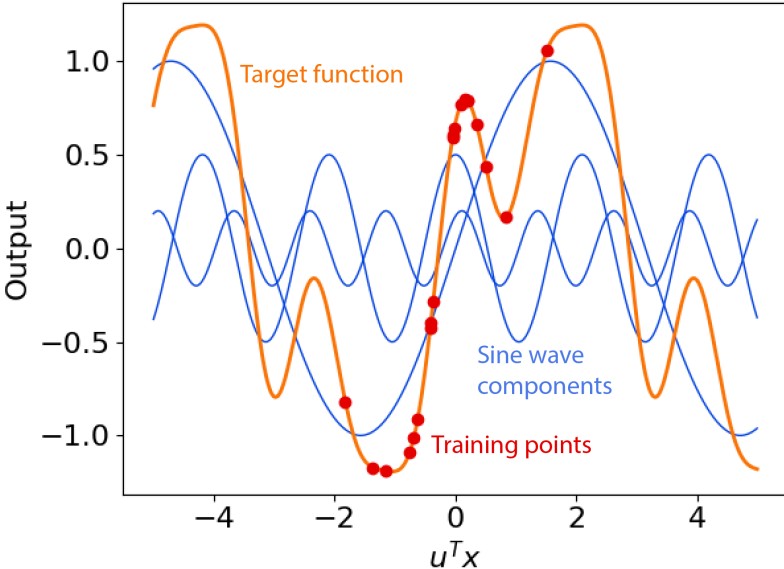

Figure 6: An illustration of a single one-dimensional projection in our sine wave regression task with sampled training points. The target function in this projection direction is made of three sine wave components summed together.

simultaneously. Fig 7 illustrates an example input. Inputs are flattened to remove all spatial structure. Outputs are $k$-hot encoded vectors constructed by concatenating the 1-hot encoded labels for each component image; thus, targets are $10k$ dimensional. For this task, accuracies are reported on average over component images (for instance, if a model correctly guesses the class of two out of four images, the accuracy would be $50\%$).

Training and test sets for this task are constructed respectively as follows: each input in the training (or test) set is produced by randomly selecting $k$ images without replacement from the original CIFAR-10 training (or test) set and concatenating them in a random permutation. With $k$ images, there are $10^k$ possible class permutations for each input. We use a fixed training set size of $10^6$; thus, the probability of a test set point having the same class permutation as a training set point is at most $\frac{10^6}{10^k}$. For large $k$, we expect each test set point to test a class permutation unobserved in the training set.

Note that this task fits the modular structure of Equation 13: $y_j(x; U_j)$ is a 1-hot encoded label of dimensionality $10k$ indicating both the label of the $j$-th component image and which of the $k$ images is being predicted by the module. The full output is constructed as a sum of the $k$ labels $y_j(x; U_j)$. As with the sine wave regression task, by increasing the number of component images, we may test generalization in arbitrarily high dimensions.

**Class combination experiments: Compositional CIFAR-10**   We consider a Compositional CIFAR-10 variant in which the training inputs are constructed to have a distinct set of class label combinations compared to the test inputs (e.g. with $k = 3$, if any training set input has the class combination cat, airplane, ship, then this class combination is not permitted on *any* test set input). This is done by partitioning the full set of class combinations into a set allocated for the training inputs and another disjoint set allocated for the test inputs. Thus, this tests out-of-distribution generalization. All other dataset parameters are set the same way as in the original Compositional CIFAR-10 task.

**Noisy inputs experiment: Compositional CIFAR-10**   We consider a Compositional CIFAR-10 variant in which the training inputs have added Gaussian noise drawn from $\mathcal{N}(0, \sigma I)$ of varying magnitude $\sigma$; this is done after the concatenation of $k$ images together. Test inputs *do not* have any added noise added. All other dataset settings are identical to the original Compositional CIFAR-10 task.

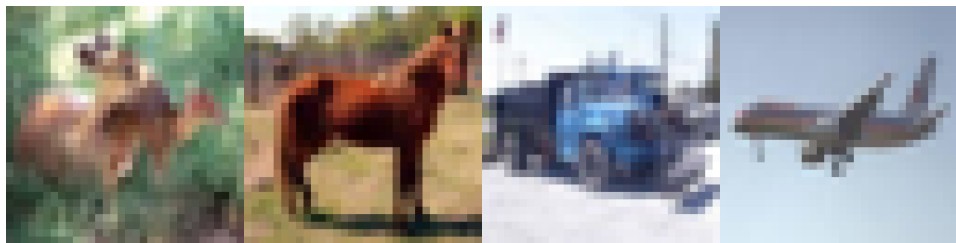

Figure 7: An example input from the Compositional CIFAR-10 task with 4 component images. The goal of the task is to predict the classes of all component images in the input. In this case, with 4 component images, the output target would be a 4-hot encoded 40-dimensional vector representing the true class of each of the 10 possible classes for each component image.

### E.2 EXPERIMENTS ON MONOLITHIC NETWORKS

**Architecture and hyperparameter settings: sine wave regression**  In our experiments, all neural networks are fully connected and use ReLU activations except at the final layer. We do not use additional operations in the network such as batch normalization. Networks are trained using Adam (Kingma & Ba, 2015) to minimize a mean squared error loss. We perform a sweep over learning rates in $\{0.001, 0.01, 0.1\}$ and find that the learning rate of $0.01$ performs best in general over all experiments, as justified by Fig 8 in Appendix E. Our results are reported in this setting. All networks are trained for 10000 iterations which we find to be generally sufficient for convergence of the training loss.

The network architectures are varied as follows: the width of the hidden layers is selected from $\{8, 32, 128\}$, and the number of layers is selected from $\{3, 5, 7\}$. This yields 9 total architectures. In order to consistently measure the number of parameters for an architecture as the input dimensionality varies, when we count the number of parameters we treat the input dimensionality $m$ as fixed at $m = 1$. Note that this slightly underestimates the true number of parameters in each NN. The values of $k$ and $m$ range from 2 to 9. The value of $n$ ranges from 100 to 100000.

All experiments are run over 5 random seeds and results are averaged. Experiments are run on a computing cluster with GPUs ranging in memory size from 11 GB to 80 GB.

**Fitting our theoretical model: sine wave regression task**  Our theoretical model of generalization has three free parameters $c$, $\Omega$ and $\alpha$. We select these parameters to best match the empirically observed trends of training set performance on the sine wave regression task; we find that $c = 1.15$, $\Omega = 1.57$, and $\alpha = 0.85$.

**Architecture and hyperparameter settings: Compositional CIFAR-10**  In our experiments, all neural networks are fully connected and use ReLU activations except at the final layer. Note that the inputs are flattened; our networks do not use the spatial structure of the input. Batch normalization is applied before each ReLU. Images are normalized with standard normalization.

Networks are trained using Adam (Kingma & Ba, 2015) to minimize softmax cross-entropy loss using a learning rate of 0.0001 and batch size of 128. Note that the loss is averaged over all component images for each input. All networks are trained for a single epoch on 1000000 random training points; note that with a moderately large number of images in each input, the total number of possible training inputs can be much larger. We use a test set of size 10000.

The network architecture consists of fully connected layers of size 512, 512, 512, 256 and 128 before a final fully connected layer to predict the output label.

Experiments are run over five random seeds for each hyperparameter configuration. Experiments are run on a computing cluster with GPUs ranging in memory size from 11 GB to 80 GB.

### E.3 EXPERIMENTS ON MODULAR NETWORKS

---

**Algorithm 1** Finding a single module projection $\hat{U}$

---

**Require:** Supervised training set $(X, y(X))$, iterations $iters$, number of training points $n$, learning rate $\eta$, batch size $b$, kernel function $\kappa$
  Randomly initialize $\hat{U}$
  **for** $iter = 1, \ldots, iters$ **do**
    Initialize $b \times b$ kernel matrix $\mathbf{K}$
    Randomly subsample $b$ training points from $X$ and store in $\tilde{X}$
    **for** $i = 1, \ldots, b$ **do**
      **for** $j = 1, \ldots, b$ **do**
        $\mathbf{K}[i, j] = \kappa(\tilde{x}_i, \tilde{x}_j; \hat{U})$; index points from $\tilde{X}$
      **end for**
    **end for**
    $\mathcal{L} = y(\tilde{X})^T \mathbf{K}^{-1} y(\tilde{X})$
    Compute $g_{\hat{U}} = \nabla_{\hat{U}} \mathcal{L}$; can be found with automatic differentiation
    $\hat{U} = \hat{U} - \eta g_{\hat{U}}$
  **end for**
  Return $\hat{U}$

---

**Algorithm 2** Binary search for sample complexity

---

**Require:** A training algorithm $\mathcal{T}(n)$ which outputs test loss of model when trained on $n$ samples; the desired error $\epsilon$; the number of binary search iterations $B$.
  Initialize $l = 0$ and $r = \infty$: our current guess for the number of required samples lies in $[2^l, 2^r]$.
  Initialize $c = 12$: our current guess for the number of required samples is $2^c$.
  **for** $b = 1, \ldots, B$ **do**
    Find test loss $e = \mathcal{T}(2^c)$
    **if** $e > \epsilon$ **then**
      Increase number of samples
      $l = c$
      **if** $r = \infty$ **then**
        $c = c + 2$
      **else**
        $c = (c + r)/2$
      **end if**
    **else**
      Decrease number of samples
      $r = c$
      $c = (l + c)/2$
    **end if**
  **end for**
  Return $2^c$

---

**Architecture and hyperparameter settings: sine wave regression task**    Each module $\hat{y}_j(x; \hat{U}_j)$ is constructed as follows: $\hat{U}_j$ consists of two components $\hat{u}_j$ and $\hat{v}_j$. The module output is constructed as:

$$\hat{y}_j(x; \hat{U}_j) = f_j(\hat{u}_j^T x) \tag{72}$$

where $f_j$ represents a neural network with scalar input and output.

We set the kernel $\kappa$ as follows:

$$\kappa(x_1, x_2; (\hat{u}, \hat{v})) = e^{-\frac{1}{2\sigma^2}\left(\frac{x_1^T \hat{u}}{\hat{v}^T \hat{u}} - \frac{x_2^T \hat{u}}{\hat{v}^T \hat{u}}\right)^2}$$
$$+ e^{-\frac{1}{2\sigma^2}\left(x_1 - \frac{x_1^T \hat{u}}{\hat{v}^T \hat{u}}\hat{v} - x_2 + \frac{x_2^T \hat{u}}{\hat{v}^T \hat{u}}\hat{v}\right)^2 \hat{v}} \tag{73}$$

where $\sigma$ is a hyperparameter. Intuitively, $\hat{v} \in \mathbb{R}^m$ corresponds to a direction along which projection directions $\hat{u}$ of other modules are not sensitive. This choice of kernel is motivated by the observation that if $\hat{v}^T \hat{u}_j$ for $j \neq i$, then $\left(x - \frac{x^T \hat{u}_i}{\hat{v}^T \hat{u}_i}\hat{v}\right)^T \hat{u}_j = x^T \hat{u}_j$ and $\left(x - \frac{x^T \hat{u}_i}{\hat{v}^T \hat{u}_i}\hat{v}\right)^T \hat{u}_i = 0$: $x - \frac{x^T \hat{u}_i}{\hat{v}^T \hat{u}_i}\hat{v}$ removes all the information in $x$ relevant to module $u_i$ while retaining information relevant to all other modules.

Due to the computational cost of computing sample complexity via binary search (Algorithm 2), we fix several hyperparameters by small-scale experiments before the final experiment to control the total runtime. We first sweep through some parameters of in our module initialization method (Algorithm 1). Specifically, we perform sweep over module batch size in $\{32, 128, 512\}$, module learning rate in $\{0.001, 0.01, 0.1\}$, and module iteration number in $\{100, 1000\}$. The combination of module batch size 128, module learning rate 0.1, and 1000 module iterations has the lowest test error in our experiment.

Then, we sweep through number of architectural modules in $\{1, k, 2 \times k, 5 \times k\}$, learning rate in $\{0.001, 0.01, 0.1\}$ and $\sigma$ in $\{0.3, 1.0, 3.0\}$. We find that the combination of module number $5 \times k$, learning rate $= 0.01$ and $\sigma = 1.0$ achieves best test performance. In addition, for our main experiments on modular NNs, all networks for dimension ($k = m$) smaller than 7 are trained for 1000 iterations, while dimension 7 and 8 networks are trained for 700 iterations, dimension 9 networks are trained for 500 iterations and dimension 10 networks are trained for 200 iterations. The high-dimension networks are stopped early since a smaller number of iterations was sufficient to converge on the training set; these numbers are determined based on small-scale experimental observations.

In binary search, we stop the search when the higher bound and the lower bound are close enough ($r - l < 0.3$) to shorten our runtime. Also, due to GPU memory limitations, we can only support a sample size up to $10^6$, so we stop our experiments when the current sample size reaches $2^{22}$ ($c = 22$). We set the maximum search iteration ($B$) to be 18.

We test our network in 9 network architectures: the width of the hidden layers is selected from $\{8, 32, 128\}$, and the number of layers is selected from $\{2, 4, 6\}$. We also have three different values ($\{0.5, 1.0, 1.5\}$) for the desired test error $\epsilon$ in binary search so as to pinpoint the most suitable value for further application of our method; we select a desired error of 1.5 for our results. We keep $k = m$ in all experiments and the values range from 2 to 9. All experiments are run over 5 random seeds and on the same computing cluster described in the previous section.

**Disentanglement experiments: sine wave regression task**    For our disentanglement experiments evaluating whether modular NNs can find the true modules underlying the task, we use the following hyperparameter settings: $m = k = 10, n = 1000$. We use a modular NN with 20 modules. Each module uses a fully connected architecture with 5 layers and a hidden layer width of 32. We train the modular NN with a learning rate of 0.001 for 1000 iterations.

For learning our module initialization, we use 100 iteration steps with a learning rate of 0.01 and a batch size of 128. $\sigma$ is set to 1.0.

For constructing a t-SNE embedding, we use a perplexity of 5. For computing similarity scores, we first compute the absolute value of the cosine similarity between each pair $(U, \hat{U})$. of learned and target module directions. For each learned module $\hat{U}$, we then find the target module with the largest absolute cosine similarity. Finally, we average the maximum absolute cosine similarities across all modules to produce a similarity score: $\sum_{i=1}^{K} \max_{j=1}^{k} \frac{\hat{U}_i^T U_j}{||\hat{U}_i||_2 ||U_j||_2}$.

**Ablation experiments: sine wave regression task**    For our ablation experiments, we train on 10000 points. $k = m$ is varied from 2 to 9 and the number of architectural modules is set to $5 \times k$. Each module uses a fully connected architecture with 5 layers and a hidden layer width of 32. We train the modular NN with a learning rate of 0.001 for 1000 iterations.

For learning our module initialization, we use 100 iteration steps with a learning rate of 0.01 and a batch size of 128. $\sigma$ is set to 1.0.

**Architecture and hyperparameter settings: nonlinear sine wave regression task**    To learn the nonlinear sine wave regression function, we consider modular architectures of the form:

$\frac{1}{\sqrt{K}} \sum_{j=1}^{K} \hat{y}_j(||\hat{U}_j - x||_2)$ where $\hat{y}_j$ is a neural network and $\hat{U}_j$ are learned parameters. We apply our method to learn an initialization for modules $\hat{U}_j$ using the following kernel:

$$\kappa(x_1, x_2; \hat{U}_i) = e^{-\frac{1}{2\sigma^2}(||x_1 - \hat{U}_i|| - ||x_2 - \hat{U}_i||)^2} \tag{74}$$

We consider modules constructed as fully connected networks with 6 layers and width 128. We set $k = m = 5$ and set $n = 1000$. All other hyperparameter settings are consistent with our original sine wave regression experiments.

**Architecture and hyperparameter settings: Compositional CIFAR-10**     Note that given an input composed of $k$ images, the flattened input dimensionality is $3072k$. Each module $\hat{y}_j(x; \hat{U}_j)$ is constructed as follows: $\hat{U}_j \in \mathbb{R}^{3072k \times 512}$ and the module output is constructed as:

$$\hat{y}_j(x; \hat{U}_j) = f_j(\hat{U}_j^T x) \tag{75}$$

where $f_j$ represents a neural network with a $512$ dimensional input and an output of dimension $10k$.

We set the kernel $\kappa$ as follows:

$$\kappa(x_1, x_2; \hat{U}) = e^{-\frac{1}{2\sigma^2}||x_1^T \hat{U} - x_2^T \hat{U}||_2^2} I \tag{76}$$

where $\sigma$ is a hyperparameter; we set $\sigma = 20.0$ to match the scale of the distances between projected inputs. To make kernel optimization more efficient, we stochastically optimize only the components of $y(X)^T \mathbf{K}^{-1} y(X)$ corresponding to a single class at a time.

Unless otherwise specified, hyperparameters are set to be consistent with the monolithic network. Additional hyperparameters are set as module batch size $128$ and module learning rate $0.01$. Module optimization is performed over a single pass over the training data. We fix the number of architectural modules as 32. Each module is a ReLU-activated neural network with hidden layer sizes 256, 128 and 64. Each ReLU is preceded by batch normalization. The module outputs are all concatenated and fed into a final linear layer to produce the $10k$ dimensional output. Critically, *all the modules have the same weights*. This is done to match the properties of the task: each modular component of the task is the same, namely to predict the class of a single CIFAR-10 image. This is unlike the sine wave regression task, where each modular component corresponds to a different function.

We vary the number of images $k$ from 1 to 8. All experiments are run on 5 random seeds and on the same computing cluster described in the previous section.

# F  ADDITIONAL EXPERIMENTS

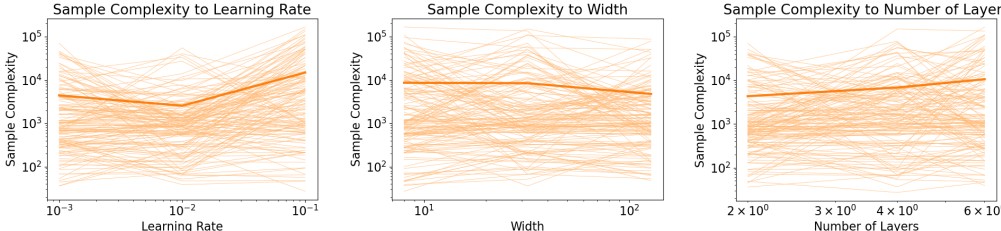

Figure 8: Performance of monolithic architecture on a sine wave regression task across learning rates of $\{0.001, 0.01, 0.1\}$ (left), widths of $\{8, 32, 128\}$ (center), and number of layers $\{3, 5, 7\}$ (right). Each line represents a different architecture and desired test error, and the average performance is shown in bold. Note in the left plot that a learning rate of $0.01$ tends to perform the best, so all our experiments are reported in this setting.

Table 3: Comparison of our method with baselines on a Compositional CIFAR-10 variant in which the training inputs are constructed to have a distinct set of class label combinations compared to the test inputs (e.g., if any training set input has the class combination cat, airplane, ship, then this class combination is not permitted on any test set input). Thus, this tests combinatorial out-of-distribution generalization. We find test set accuracies for inputs with 6 component images. Standard errors over 5 trials are reported.

| Method | Test Accuracy |
|---|---|
| Baseline monolithic | $42.56\% \pm 0.07\%$ |
| Baseline modular | $45.26\% \pm 0.06\%$ |
| Our method | $49.90\% \pm 0.15\%$ |

Table 4: Comparison of our method with baselines on a Compositional CIFAR-10 variant in which the training inputs have added Gaussian noise drawn from $\mathcal{N}(0, \sigma I)$ of varying magnitude $\sigma$. We find test set accuracies for inputs with 6 component images (note that the test points do not have added noise). This tests out-of-distribution generalization to small distribution shifts. Standard errors over 4 trials are reported.

| Noise Level | Baseline Monolithic | Baseline Modular | Our Method |
|---|---|---|---|
| 0 | $42.92 \pm 0.05\%$ | $45.66 \pm 0.14\%$ | $50.49 \pm 0.09\%$ |
| 0.3 | $42.57 \pm 0.05\%$ | $45.42 \pm 0.10\%$ | $50.27 \pm 0.08\%$ |
| 1 | $39.47 \pm 0.09\%$ | $43.07 \pm 0.08\%$ | $46.95 \pm 0.11\%$ |
| 3 | $31.75 \pm 0.03\%$ | $34.71 \pm 0.05\%$ | $34.67 \pm 0.13\%$ |
| 10 | $21.76 \pm 0.10\%$ | $24.31 \pm 0.34\%$ | $24.41 \pm 0.13\%$ |
| 30 | $11.09 \pm 0.33\%$ | $12.50 \pm 0.55\%$ | $12.29 \pm 0.40\%$ |

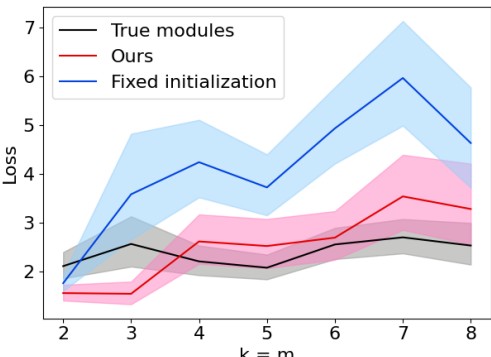

Figure 9: Test loss of various training methods of training a modular NN on a sine wave regression task as the dimensionality of the task $k = m$ is varied. Margins indicate standard deviations over $5$ random seeds. Ours indicates that NN module directions $\hat{U}$ are initialized using our method and then trained on the task. Fixed initialization indicates that module directions are learned with our method and are fixed during task training. True modules indicates that $\hat{U}$ is set to the underlying module directions of the task $u$ (which are generally unknown) and fixed during task training. Observe that when $k = m$ is large, using the true, ground-truth module directions slightly outperforms our method, although interestingly our method performs better for low-dimensional tasks. Fixing the initialization found by our method results in significantly worse performance relative to allowing the module directions to vary.

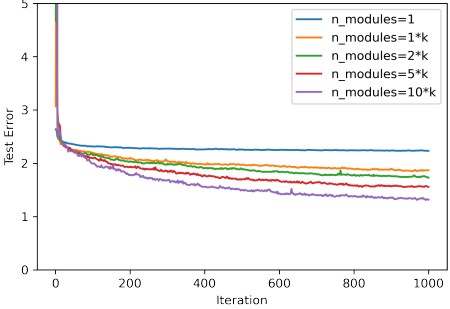

Figure 10: Test loss of a modular NN initialized with our method with different numbers of modules on a sine wave regression task. The experiment is conducted with all hyperparameters fixed at the optimal value detailed in App E except the learning rate and $\sigma$. The experiment is run under 5 random seeds. The lines are averaged over all runs. Model performance increases as the number of modules increases. Intuitively, as the number of modules increases, the model can contain more information, thus achieving lower error.

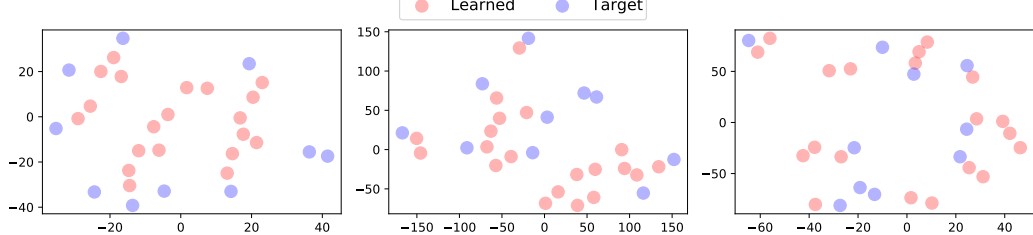

Figure 11: t-SNE embedding of target ($u$) and learned module projections ($\hat{u}$) learned by a randomly initialized modular NN without training (left), randomly initialized modular NN trained with gradient descent (center), and our initialized modules (right) on a sine wave regression task. From left to right, learned modules cluster more closely around target modules.

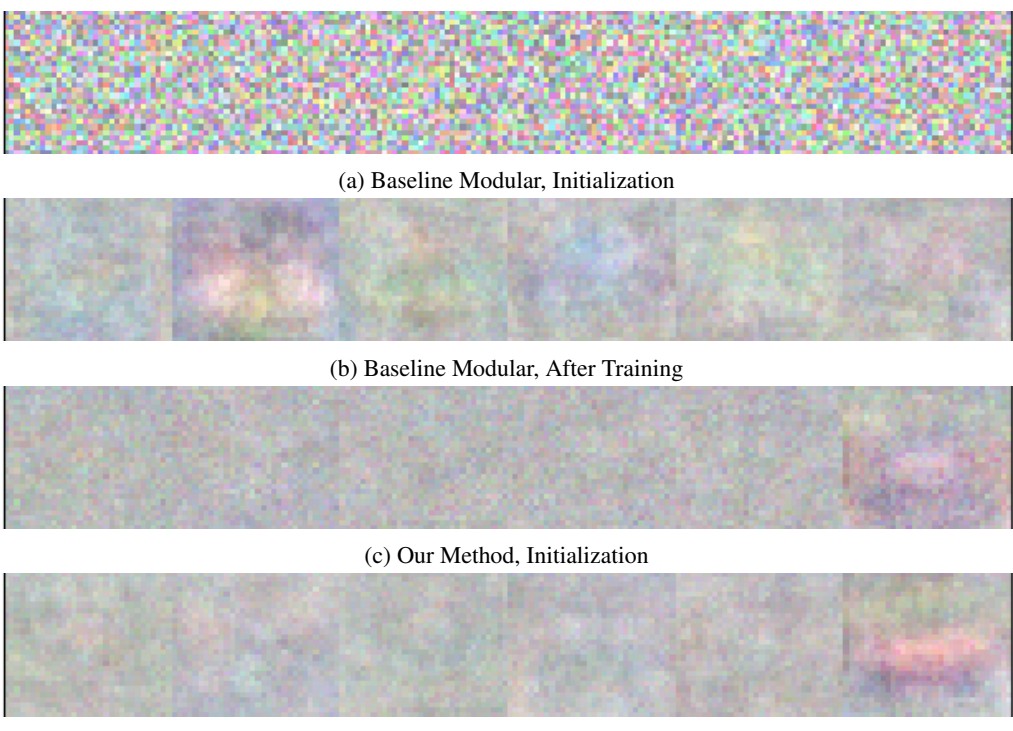

Figure 12: Illustration of the weights $\hat{U}$ before and after training using the baseline modular method and our method on the Compositional CIFAR-10 task with 6 component images. A single column of the $\hat{U}$ corresponding to a single module is plotted. The weights are organized into 6 groups corresponding to which of the component images each weight is sensitive to. Intuitively, this plots the sensitivity of a single module to each of the component images. (a) At initialization, the baseline modular method has randomly initialized $\hat{U}$. (b) After training, it learns to be mostly sensitive to a particular image in the input. (c) Our method learns to be sensitive to only a single image (the rightmost image) in the input *before any training on the task*, thus correctly learning the underlying modular structure of the task. (d) The module in our method retains its sensitivity to the original rightmost image over the course of training.

