# OpenReview forum: "Breaking Neural Network Scaling Laws with Modularity"
_ICLR.cc/2024/Conference — Submitted to ICLR 2024_

### Official Review · Reviewer_VLQY · 2023-10-30

**Soundness:** 2 fair
**Presentation:** 3 good
**Contribution:** 3 good
**Rating:** 6
**Confidence:** 3

**Summary:**

The paper seeks to understand scaling laws for modular neural networks and proposes a method for training them. Modular neural network here refers to models that sum the output of their constituent modules each of which receive (different) low-dimensional projections of the input. The paper theoretically shows that when the modules are linear neural networks that receive a linear projection of the input into a fixed dimensional space, and the data comes from a model of the same form, sample complexity is independent of the task intrinsic dimension $m$ (in contrast to the monolithic case where it is exponential in $m$). The paper then proposes a kernel-based rule to learn the initializations of the input projections from data and test the empirical performance on a sine wave regression task and compositional CIFAR.

**Strengths:**

Understanding the sample complexity of training modular vs. monolithic neural networks is an important open problem for which a theoretical contribution is potentially impactful.
The theory identifies a reasonable setting for a tractable analysis and is overall convincing (without having checked the proofs in the appendix).
Overall the paper is well presented and transparent about the merits and limitations of the analysis.

**Weaknesses:**

The scaling behaviour is studied theoretically in the case of linear neural networks for tractability. A more thorough empirical investigation to what extent this scaling law is practically relevant in the nonlinear setting would have been useful. As far as I understand the experiments conducted do not reflect the theoretical result of constant sample complexity in the input dimension. I was missing a discussion on this point.

I am a bit worried about the reproducibility of the empirical part of the paper since no code was provided as part of the submission. I also encourage the authors to specify the exact number of seeds per experiment in Figure 3b as "up to five seeds" as stated in the caption could technically mean only one seed is reported.

**Questions:**

1. The modular learning rule minimizing the norm of the $\theta_i$ is applied as a pretraining step assuming that the $\varphi(X;\hat{U}_i$ are sufficiently expressive. Since this is before training, can you elaborate why this assumption might be justified and to what extent the algorithm is robust to a violation of it?
2. There are discrepancies between the theory and toy model in Figure 5 as the paper points out in App A.2. Can you elaborate why this is not a matter of concern for the theory, i.e. what exactly causes the mismatch?
3. Figure 5 is missing labels and the caption is a bit sparse. Could you specify how exactly the four plots differ? Maybe adding a colour bar to indicate the values of the light lines could be helpful? How do the theoretical predictions look like for individual (light) lines?
4. Figure 12 is missing a legend for what the colours encode. Could you please clarify?

Suggestions / typos:
- I think it would be useful to show both the theoretical prediction and empirical validation in Figure 2 (similar to Figure 5 in Appendix A).
- Page 7 "and the test loss and dependence of the test loss"

---

> ### Author Response · Authors · 2023-11-18
> **Response Part 1/2**
>
> Thank you for your valuable and constructive comments and suggestions. We are glad to hear of your positive impression of our work.
>
> **Practical relevance of scaling law in nonlinear setting**
>
> We first would like to clarify that the empirical validation of Section 3.3 is performed on nonlinear neural networks; thus, the generalization error of linear models can predict generalization in *nonlinear* networks.
>
> We also note that although we primarily consider nonlinear model architectures containing linear module projections (as in Eqn 3), in Section 4.3 Table 2, we also conduct experiments on nonlinear module projections. As we discuss in Section 5, we believe that further studying nonlinear module projections is an important direction for future work, particularly in practical settings where nonlinear module projections may be more realistic.
>
> **Discrepancy with theory for sample complexity trend**
>
> Indeed, as the reviewer notes, while the theory predicts that in modular architectures, sample complexity scales as a constant with input dimensionality, our empirical results indicate that sample complexity grows with input dimensionality. We believe the discrepancy with the theory arises due to the optimization challenges associated with modular architectures (as previously found in Csordas et al. and Mittal et al.). Specifically, with higher dimensional inputs, the gap between the performance of the optimal modular model and of the modular model learned by gradient descent grows. Our modular learning rule introduced in Section 4.2 is aimed at easing these optimization difficulties and is effective as illustrated in Figure 3. However, it still does not find the optimal modular architecture parameters, thus maintaining an input dimensionality dependence for sample complexity. We regard further improving the optimization of modular architectures as an important future direction.
>
> We have added a discussion of this point in our revised Section 5.
>
> **Code**
>
> Thank you for the suggestion. In our latest revision, we have included code to reproduce the empirical validation of Section 3.3 and the empirical results of Section 4.3.
>
> **Number of seeds**
>
> Due to experimental limitations at the time of submission, we were unable to run 5 seeds for each of the experimental settings for Compositional CIFAR-10. We have now updated our results with all 5 random seeds for each setting. Note that the error bars in Figure 3 indicate standard errors individually for each point.
>
> **Sufficient expressivity assumption**
>
> Thank you for highlighting this important point: indeed, we assume the features of Equation 18 are sufficiently expressive in the sense that $pK > dn$. Effectively, this assumption states that for any choice of module projections $\hat U_i$, some set of module parameters can perfectly fit the training data. This assumption is natural when modules are neural networks since they are typically capable of fitting a range of training datasets. When this assumption does not hold, the pseudoinverse solution of Equation 19 finds a solution minimizing the mean squared error between the predicted and actual training targets $y(X)$. In this case, the training set prediction error is generally not $0$ (except when $\hat U_i$ are at their correct values). Importantly, minimizing the norm of $\theta$ with respect to $\hat U_i$ *may not necessarily* yield a lower prediction error. Thus, we do not necessarily expect our algorithm to be effective when modules are not sufficiently expressive.
>
> We have added a discussion of this point in our revised Section 4.2.

---

> ### Author Response · Authors · 2023-11-18
> **Response Part 2/2**
>
> **Discrepancy in Figure 5**
>
> We note two key discrepancies between our theory and empirical results: first, the loss is empirically larger than predicted for small amounts of training data, and second, the error spike at the interpolation threshold is smaller than predicted by the theory.
>
> We believe the first discrepancy is due to imperfect optimization of neural networks, especially in low data regimes. Note that the linearized analysis assumes that the linear model solution finds the exact global optimum. However, the actual optimization landscape for modular architectures is highly non-convex, and the global optimum may not be found especially for small datasets (indeed, we find a significant discrepancy between predicted and actual training loss values for small data size n; in the overparameterized regime, the predicted training error is exactly $0$). We believe this causes the discrepancy between predicted and actual test error in low data regimes.
>
> We hypothesize that the second discrepancy is also partly due to imperfect optimization. This is because the interpolation threshold spike can be viewed as highly adverse fitting to spurious training set patterns. This imperfect optimization is more pronounced at smaller $m$. Despite these discrepancies, we nevertheless find that our theory precisely captures the key trends of empirical test error.
>
> We have added a more thorough discussion of this point in our revision.
>
> **Comments on Figure 5**
>
> Thank you for the suggestion. In response, we have edited the caption of Figure 5 and modified the Figure to indicate the meaning of the light vs dark lines as well as indicate the difference between the light lines. We do not include the theoretically predicted curves for the light lines for visual ease of understanding; however, for larger $m$, the predicted test loss is higher while the location of the interpolation threshold is retained, and for larger $p$ the predicted test loss shifts vertically up and down.
>
> The four plots indicate trends of training and test loss along four different dimensions: $k$ (number of modules), $m$ (input dimensionality), $p$ (model size) and $n$ (training set size).
>
>
> **Figure 12 clarification**
>
> Note that the Compositional CIFAR-10 inputs have shape 32x32kx3. In our modular architecture, the module input dimensionality is 512 (see Appendix E.3). Therefore, each learned input projection $\hat{U}$ each has shape 32x32kx3x512. In Figure 12, we plot a representative slice of $\hat{U}$ with shape 32x32kx3; note that since it has the same shape as the original image, we may plot it as an image with channel dimensions encoding RGB values.
>
> We would be happy to clarify this in our revision if necessary.
>
> **Empirical result on Figure 2**
>
> Thank you for this suggestion. We are unable to use our current empirical results to generate an empirical sample complexity curve since our current experiments in this setting compute the test set error at a given pair of training set size n and input dimensionality m. However, we are currently running binary search over training set size n to compute the minimum n required to achieve the desired test set error of 1.2. We will update the figure with empirical results once available.
>
> **Page 7 typo**
>
> Thank you for pointing this out; we have corrected this in our revision.

---

> > ### Comment · Reviewer_VLQY · 2023-11-21
> > **Response to rebuttal**
> >
> > Thank you for your thorough response to my review that has helped clarify my questions. I appreciate that you now provide the code to reproduce the empirical verification. Conditioned on the promised addition of the currently missing seeds and given no corresponding surprises as a result (I find it worrying that some unspecified part of the result relies on a single seed), I maintain my score for acceptance.

---

> > > ### Author Response · Authors · 2023-11-23
> > >
> > > Thank you for your follow-up; we are glad to hear our response was clarifying.
> > >
> > > As requested, we have finished running all the missing seeds and have updated our results: the revised Figure 3b now includes results for 5 random seeds for each setting.

---

### Official Review · Reviewer_MGmV · 2023-10-31

**Soundness:** 3 good
**Presentation:** 3 good
**Contribution:** 3 good
**Rating:** 5
**Confidence:** 2

**Summary:**

This paper presents a theoretical model of NN learning, specifically predicts that while the sample complexity of non-modular NNs varies exponentially with task dimension, sample complexity of modular NNs is independent of task dimension. The authors then develop a learning rule to align NN modules to modules underlying high-dimensional modular tasks, and presents empirical results which demonstrate improved performance of modular learning.

**Strengths:**

The paper presents the first theoretical model to explicitly compute non-asymptotic expressions for generalization error in modular architectures, develops new modular learning rules based on the theory and empirically demonstrated the improved performance of the new method.

**Weaknesses:**

Validation of theoretical results is only shown in the appendix, with large discrepancy between theoretical predictions and numerics, I think more empirical evaluations are needed to verify the theoretical result.

**Questions:**

1. What causes the large deviation of the test loss between actual and predicted in Figure 5?
2. In figure 4 (also figure 3b), the total range of the similarity score is quite small, it is therefore difficult to say whether the result is a significant improvement from baseline.

---

> ### Author Response · Authors · 2023-11-18
>
> Thank you for your valuable comments.
>
> **Concerns on the discrepancy between actual and predicted test loss**
>
> First, we'd like to note that the prediction of NN test loss in Figure 5 is relatively strong considering that we are using a *linear* model to approximate the generalization trends of a highly *nonlinear* neural network. Importantly, we are able to predict the location of the interpolation threshold, and we believe the results are overall competitive with prior work modeling NN generalization.
>
> We note two key discrepancies between our theory and empirical results: first, the error spike at the interpolation threshold is smaller than predicted by the theory, and second, the loss is empirically larger than predicted for small amounts of training data. We believe the discrepancies are due to imperfect optimization, especially in regimes of small training data and small model size. Note that the linearized analysis assumes that the linear model solution finds the exact global optimum. However, in the actual optimization landscape for modular architectures is highly non-convex, and the global optimum may not be found especially for small models and datasets (indeed, in Figure 5, note the discrepancy between predicted and actual training loss values for small model size p and small data size n). We believe this causes the large discrepancy between predicted and actual test error in low data regimes. Moreover, we believe this imperfect optimization leads to a smaller-than-expected test error at the interpolation threshold given that the interpolation threshold spike can be viewed as highly adverse fitting to spurious training set patterns.
>
> We have added a more thorough discussion of this point in our revised Section 3.3.
>
> **Significance of improvement over baseline**
>
> Regarding Figure 3b, note that results are averaged over 5 seeds and the standard error margins plotted indicate that the improvement over the baseline is statistically significant. Note that for the highest number of images tested, the improvement in accuracy is roughly 5% which we believe is a significant improvement.
>
> Regarding Figure 4, we believe that while the difference in raw similarity score between our method and the baseline appears small, this corresponds to a large difference in the qualitative behavior of the two networks. To illustrate this, in Figure 12 we have plotted an illustration of a learned module direction under our method relative to the baseline. We find that the module direction learned by our method is visibly sensitive to only one component image while the baseline is not; thus, our method learns the underlying module structure of the task.

---

### Official Review · Reviewer_Jw6v · 2023-11-01

**Soundness:** 4 excellent
**Presentation:** 4 excellent
**Contribution:** 3 good
**Rating:** 8
**Confidence:** 3

**Summary:**

This paper analyzes the sample complexity of modular neural networks and shows theoretically how the sample complexity of modular networks doesn't depend on the intrinsic dimensionality of the input. This is proven for linear models. The theory is supported by experiments on 1) sin wave regression and 2) compositional CIFAR10. The paper further proposes a learning rule to ensure the modularity of the task is aligned with the modularity of the network.

**Strengths:**

1. This is the first paper to conduct a rigorous theoretical analysis of modular neural networks. Understanding the empirical success of modular neural networks is an important open problem.

2. The theoretic analysis and the effect of different terms in the generalization bound are presented clearly.

3. Assumptions for the theoretical analysis are presented clearly.

4. Related work is covered well and in thorough detail.

**Weaknesses:**

1. Including synthetic experiments in the linear model to demonstrate how the sample complexity changes for modular and non-modular networks in a specific setting.

**Questions:**

N/A

---

> ### Author Response · Authors · 2023-11-18
>
> Thank you for your valuable comments. We are glad to hear of your positive impression of our work.
>
> **Comments on synthetic experiments on linear model**
>
> As the reviewer notes, our theoretical analysis hinges on modeling *nonlinear* neural networks as linear models. However, our experiments are conducted on nonlinear networks and tasks. This is because our theoretical analysis is aimed at capturing generalization trends in actual, nonlinear neural networks, and can do so accurately as demonstrated in Appendix A.2.
>
> Moreover, our construction of modular tasks and models (in Equations 3 and 6) is *fundamentally nonlinear*: the input is linearly projected, fed through nonlinear functions, and then summed. The linearization analysis of Section 4.1 separately linearizes the network with respect to the projections and the parameters of the nonlinear functions; this separation of parameters ultimately yields a superior sample complexity scaling for modular networks relative to monolithic networks. However, for linear models, there is no corresponding notion of notion of modularity: fundamentally, there is no division of parameters into "projection" parameters and "module" parameters. We believe our linearization analysis is relevant only for nonlinear models.
>
> Thus, while we understand the spirit of the reviewer's suggestion, we are unclear on what exact experiment would be most valuable to run. However, we would be happy to include any specific experiments suggested by the reviewer.

---

### Meta-Review · Area_Chair_f2tM · 2023-12-12

**Metareview:**

The reviews of this paper are fairly favorable, for instance, Reviewer Jw6v highlights the value of rigorous theory for modular networks and the clarity of the results. While there is agreement on the importance of modularity, eventually there are two important weaknesses: The theoretical novelty of this work is rather limited and the authors do not refer to or discuss related literature on random matrix theory (RMT), which contain similar or stronger results. Here I am referring to the fundamental methodology and not necessarily to the modularity-specific problem formulation. Concretely, there is a concern that Theorem 1 is more or less  a straightforward application of existing results in RMT. In fact, literature contains a lot more sophisticated results e.g. general feature distributions rather than assuming i.i.d. N(0,1) features. To further clarify:

- The authors are considering a so-called misspecified model where they fit p out of P parameters. This model is utilized in Hastie et al. which analyzes the same setting (besides their many other results). See [Hastie et al.](https://arxiv.org/pdf/1903.08560.pdf)'s Theorem 4. It seems this essentially yields the first equation in Theorem 1.

- The second equation of Thm 1 should be a corollary of Hastie et al. by plugging in the author's power law assumption of $i^{-\alpha}$ with $\alpha=\Omega^{-m}$ (apply an integration to obtain the $trace(\Lambda_i)$ terms basically). Such power laws on both weights and feature covariance are also extensively studied in the RMT literature to develop scaling laws. For instance, see [Cui et al. A](https://proceedings.neurips.cc/paper/2021/file/543bec10c8325987595fcdc492a525f4-Paper.pdf), [Cui et al. B](https://arxiv.org/pdf/2201.12655.pdf), [Wei et al.](https://proceedings.mlr.press/v162/wei22a/wei22a.pdf), [Jin et al.](https://arxiv.org/pdf/2110.12231.pdf) etc. Again the results of these works apply under more general feature distributions. None of these works nor Hastie et al. are cited. There should have been some discussion of theory on test/train risk in light of literature.

- Finally, the proof of Thm 2 basically relies on Thm 1 and do not introduce technical innovation. This obviously makes sense because the authors set up the problem and the power law following Thm 1 in a modular fashion to convey their point on the sample complexity benefits of modularity. However, there seems to be no substantial theory contribution specific to modularity beyond making the right assumptions about the covariance's power law and plugging in Thm 1.

Overall, the decision to reject stems from the absence of discussion on directly related work and the associated lack of substantial theoretical novelty.

**Justification For Why Not Higher Score:**

The paper can be accepted based on reviewer scores. However, based on my evaluation, the authors have not discussed related work and their main theory results are basically already known (although I admit that they add their modularity spin to the problem).

**Justification For Why Not Lower Score:**

N/A

---

### Decision · Program_Chairs · 2024-01-16

Reject